# QUATRO: Query-Adaptive Trust Region Policy Optimization for LLM Fine-tuning

Doyeon Lee [1]   Eunyi Lyou [1]   Hyunsoo Cho [2]   Sookyung Kim [2]   Joonseok Lee [1]   Jaemoo Choi [1,3]

## Abstract

GRPO-style reinforcement learning (RL)-based LLM fine-tuning algorithms have recently gained popularity. Relying on heuristic trust-region approximations, however, they can lead to brittle optimization behavior, as global importance-ratio clipping and group-wise normalization fail to regulate samples whose importance ratios fall outside the clipping range. We propose *Query-Adaptive Trust-Region policy Optimization* (QUATRO), which directly enforces trust-region constraints through a principled optimization. This yields a clear and interpretable objective that enables explicit control over policy updates and stable, entropy-controlled optimization, with a stabilizer terms arising intrinsically from the exact trust-region formulation. Empirically verified on diverse mathematical reasoning benchmarks, QUATRO shows stable training under increased policy staleness and aggressive learning rates, maintaining well-controlled entropy throughout training.

## 1. Introduction

Long-horizon reasoning like math and code has been considered a bottleneck for LLMs, yet today's frontier models have ascended to expert-level performance on the most sophisticated benchmarks. This leap is fueled by reward-based fine-tuning (Guo et al., 2025; Yang et al., 2025a), which enables the active and iterative refinement of a goal-directed policy beyond passive imitation of static data. In this setting, a critical challenge is ensuring that each update improves performance without destroying the model's fundamental capabilities. Trust-region (TR) policy optimization (Schulman et al., 2015) provides a natural framework for this by explicitly limiting the magnitude of each policy update, thereby preventing overly aggressive changes while allowing consistent improvement.

This has motivated various practical RL fine-tuning algorithms that incorporate trust-region principles in simplified forms. Among them, Group Relative Policy Optimization (GRPO) (Shao et al., 2024) and Group-Scale Policy Optimization (GSPO) (Zheng et al., 2025a) have demonstrated strong empirical efficiency. They regulate policy updates through heuristic mechanisms, most notably importance-ratio clipping, instead of enforcing an explicit trust-region constraint. While empirically effective, GRPO and its variants (Shao et al., 2024; Zheng et al., 2025a; Zhao et al., 2026) exhibit two major structural limitations that stem from heuristic approximations to trust-region policy optimization.

First, GRPO is highly sensitive to optimization hyperparameters, and it often becomes unstable under *policy staleness* due to its reliance on heuristic importance-ratio clipping. To constrain radical policy updates and keep the updated policy close to the one from the previous step, GRPO employs heuristic importance-ratio clipping, when the policy abruptly changes above some predefined ratios exceed the clipping threshold. However, this mechanism induces the opposite-side issue of *gradient masking*; that is, the corresponding gradients are truncated and no longer provide corrective feedback toward the trust region (Zheng et al., 2025b; Chen et al., 2025a; Su et al., 2026).

Furthermore, existing methods apply a "one-size-fits-all" uniform trust-region constraint and deviation tolerance across queries, regardless of their intrinsic difficulty or uncertainty. Consequently, easy prompts that admit confident high-reward responses are rapidly over-amplified until they become deterministic, while complex reasoning tasks are deprived of the exploration budget they need to find correct paths. This imbalance causes probability mass to excessively concentrate on a small subset of responses (Wu et al., 2026; He et al., 2025; Bamba et al., 2025; Zheng et al., 2025b). Such a phenomenon forces the model to stop "searching" and instead fall into a repetitive rut, sacrificing the diversity of its reasoning through early deterministic be-

[1]Seoul National University [2]Department of Artificial Intelligence / Institute for Multiscale Matter and Systems, Ewha Woman University [3]Geogia Institute of Technology. Correspondence to: Hyunsoo Cho <chohyunsoo@ewha.ac.kr>, Sookyung Kim <sookim@ewha.ac.kr>, Joonseok Lee <joonseok@snu.ac.kr>, Jaemoo Choi <jchoi843@gatech.edu>.

*Proceedings of the 43rd International Conference on Machine Learning*, Seoul, South Korea. PMLR 306, 2026. Copyright 2026 by the author(s).

havior, a failure mode widely observed in current RL-based LLM fine-tuning (He et al., 2025; Peng et al., 2025; Xi et al., 2026; Wang et al., 2025b).

In this paper, we propose a principled approach, **Query-Adaptive Trust-Region policy Optimization (QUATRO)**, designed to overcome the structural instabilities of heuristic RL fine-tuning. Unlike previous methods that rely on ad-hoc clipping, we first formulate a query-conditioned trust-region problem that explicitly accounts for the heteroscedasticity of reasoning tasks. Specifically, we first formulate *a query-conditioned trust-region problem* that explicitly constrains policy updates in a manner suitable for large-scale LLM fine-tuning. From this, we derive an *exact* query-adaptive objective via a Lagrangian dual analysis, which directly enforces the trust-region (TR) constraint rather than through heuristic approximations.

By optimizing the exact TR objective, QUATRO explicitly keeps each policy update within a controlled deviation from the previous policy. This direct enforcement largely eliminates the need for importance-ratio clipping, thereby resolving the gradient masking problem and making the training process significantly more robust to off-policy staleness and learning-rate variations. Moreover, by preventing excessive amplification of high-reward trajectories in a single update, the trust-region constraint preserves exploration, thereby mitigates entropy collapse.

We conduct extensive experiments to verify the effectiveness of our proposed method, on the mathematical reasoning task as a representative playground. Our method consistently improves the performance on Qwen2.5-Math models (Yang et al., 2024), outperforming GRPO-style baselines in Pass@$k$, achieving more prominent gain with a higher $k$. Beyond accuracy, it also exhibits stable training dynamics and strong robustness to key hyperparameters, including rollout reuse and learning-rate choices. To quantify the diversity among generated solutions, we introduce *Unique Correct Count (UCC@k)* metric, directly measuring the number of valid reasoning paths recalled by the model.

We summarize our contributions as follows:

- We propose a *query-adaptive trust-region policy optimization* method that consistently improves Pass@$k$ performance over GRPO-style algorithm, with gains increasing as the sampling budget ($k$) grows.
- The resulting objective admits a *clear optimization interpretation*, providing an analyzable control knob for regulating policy updates under policy staleness.
- Our method achieves *stable optimization dynamics*, remaining robust to policy staleness and learning rate variations while maintaining controlled entropy and avoiding premature entropy collapse.

## 2. Trust-Region Policy Optimization

Originally developed in classical reinforcement learning, trust-region policy optimization plays a central role in LLM fine-tuning for reasoning (Schulman et al., 2015; 2017; Shao et al., 2024; Zheng et al., 2025a). We review the problem setup, representative approaches, and its limitations.

### 2.1. KL-Constrained Trust-Region Optimization

Let $q$ denote an input query, and let a trajectory $o = (o_1, \ldots, o_{|o|})$ be the response generated by the model. The sequence-level likelihood under a policy $\pi_\theta$ is autoregressively factorized by

$$\pi_\theta(o \mid q) = \prod_{t=1}^{|o|} \pi_\theta(o_t \mid q, o_{<t}), \qquad (1)$$

where $\pi_\theta$ denotes a stochastic autoregressive policy with learnable parameters $\theta$, and $o_{<t} := (o_1, \ldots, o_{t-1})$. A scalar reward $R(o \mid q)$ is assigned to the entire trajectory.

**Trust Region Policy Optimization (TRPO)** (Schulman et al., 2015) formulates the RL problem as the maximization of a surrogate under a KL divergence constraint, ensuring that the policy updates remain within a trust region. Formally, the TRPO problem is given by

$$\max_\theta \quad \mathbb{E}_q \mathbb{E}_{o \sim \pi_{\text{old}}(\cdot \mid q)} \left[ \frac{\pi_\theta(o \mid q)}{\pi_{\text{old}}(o \mid q)} R(o \mid q) \right], \qquad (2)$$
$$\text{s.t. } \text{KL}(\pi_{\text{old}} \parallel \pi_\theta) \leq \delta, \quad \text{or} \quad \text{KL}(\pi_\theta \parallel \pi_{\text{old}}) \leq \delta,$$

where $\delta > 0$ specifies the maximally allowed deviation between the updated policy $\pi_\theta$ and the reference policy $\pi_{\text{old}}$, measured by the expected KL divergence. The optimization in Eq. (2) is performed iteratively. At each iteration, trajectories are sampled from a fixed sampling (reference) policy $\pi_{\text{old}}$, and the target policy $\pi_\theta$ is optimized under the trust-region constraint. After each update, the reference policy is set to the updated policy, *i.e.*, $\pi_{\text{old}} \leftarrow \pi_\theta$.

**Group Relative Policy Optimization (GRPO)** (Shao et al., 2024) is a widely used algorithm for empirically optimizing KL-regularized policy objectives in LLM fine-tuning. GRPO is closely related to trust-region policy optimization in that it aims to regulate policy updates and limit deviation from a reference policy, but it achieves this through heuristic mechanisms rather than by enforcing an explicit KL-divergence constraint.

In practice, for each query or prompt $q$, GRPO samples a group of $N$ output trajectories $\{o^{(i)}\}_{i=1}^N \sim \pi_{\text{old}}(\cdot \mid q)$ from the reference (old) policy. Learning signals are constructed by comparing rewards across these rollouts, which enable efficient and scalable training. The GRPO objective is given by

$$\mathcal{L} = - \mathbb{E}_{o^{(i)} \sim \pi_{\text{old}}} \left[ \min\left( r_t^i A^i, \, \text{clip}\left( r_t^i, \, 1 - \varepsilon, \, 1 + \varepsilon \right) A^i \right) \right], \quad (3)$$

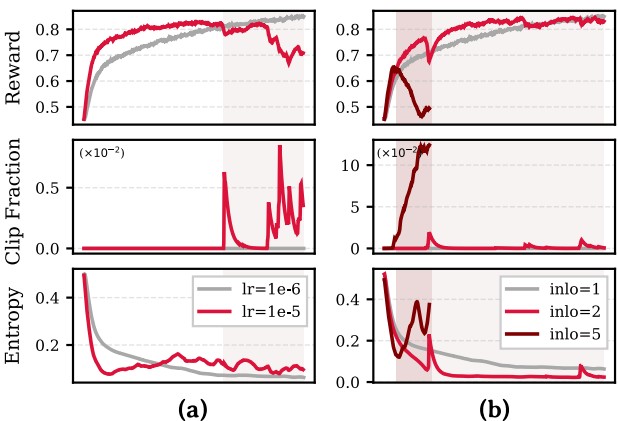

*Figure 1.* **Limitations of importance-ratio clipping of GSPO** Training dynamics under (a) increased learning rates ($10^{-6} \rightarrow 10^{-5}$) and (b) increased offline inner-loop updates (1, 2, 5), with all other settings fixed.

where $\varepsilon$ is a clipping hyperparameter to limit policy deviation, and the token-level importance ratio is defined as

$$r_t^i(o^{(i)} \mid q) = \frac{\pi_\theta(o_t^{(i)} \mid q, o_{<t}^{(i)})}{\pi_{\text{old}}(o_t^{(i)} \mid q, o_{<t}^{(i)})}. \quad (4)$$

The group-relative advantage $A_t^i$ is computed by normalizing trajectory-level rewards within the group of rollouts:

$$A^i = \frac{R_i - \text{mean}(R_1, \dots, R_N)}{\text{std}(R_1, \dots, R_N)}, \quad (5)$$

where $R_i = R(o^{(i)} \mid q)$. This group-based normalization removes the need for explicit value estimation and provides variance reduction in large-scale training.

**Grouped Sequence Policy Optimization (GSPO)** (Zheng et al., 2025a) modifies the token-level importance ratio with a sequence-level importance ratio to better align with sequence-level rewards. Specifically, GSPO defines a sequence-level importance ratio of the form

$$s^i(o^{(i)} \mid q) = \left( \frac{\pi_\theta(o^{(i)} \mid q)}{\pi_{\text{old}}(o^{(i)} \mid q)} \right)^{\frac{1}{|o^{(i)}|}}$$
$$= \exp \left( \frac{1}{|o^{(i)}|} \sum_{t=1}^{|o^{(i)}|} \log \frac{\pi_\theta(o_t^{(i)} \mid q, o_{<t}^{(i)})}{\pi_{\text{old}}(o_t^{(i)} \mid q, o_{<t}^{(i)})} \right).$$

By substituting $s^i$ into $r_t^i$ in Eq. (3), we obtain GSPO loss:

$$\mathcal{L} = - \mathop{\mathbb{E}}_{o^{(i)} \sim \pi_{\text{old}}} \left[ \min \left( s^i A^i, \text{clip}(s^i, 1 - \varepsilon, 1 + \varepsilon) A^i \right) \right]. \quad (6)$$

Despite improved alignment with sequence-level rewards, GSPO continues to rely on heuristic clipping and does not explicitly enforce a KL constraint in Eq. (2).

## 2.2. Limitations of Existing Trust-Region Methods

**Heuristic Importance-ratio Clipping.** Many GRPO-style RL fine-tuning methods (Schulman et al., 2017; Yang et al., 2025b; Ren et al., 2026; Yu et al., 2025) attempt to regulate policy updates through heuristic importance-ratio clipping (Eq. (3)). In these methods, policy updates are governed by a clipped surrogate objective with a fixed threshold $\varepsilon$, such that the importance ratio is truncated whenever it exceeds the prescribed range (*i.e.*, $s^i(o \mid q) \notin [1 - \varepsilon, 1 + \varepsilon]$). In this regime, the loss no longer varies with the actual value of $s^i(o \mid q)$, meaning that deviations beyond the clipping range are ignored during optimization. As a result, importance-ratio clipping does not control policy updates once the threshold is exceeded and does not capture how far the current policy has drifted from the sampling policy.

**Sensitivity to Hyperparameters and Policy Staleness.** As discussed, samples with importance ratios outside the clipping range escape regulation, leading to training instability. Such out-of-range samples become more frequent as *policy staleness* increases, where staleness refers to the mismatch between the current policy $\pi_\theta$ and the sampling policy $\pi_{\text{old}}$. In practice, policy staleness can arise in two common scenarios: 1) using a large learning rate, which promotes larger policy updates, and 2) reusing rollout samples, which increases the discrepancy between the sampling and update policies. To empirically validate this behavior, we evaluate GSPO under both conditions. Specifically, we 1) increase the learning rate and 2) increase the number of optimization steps performed on each sampled response, which we refer to as the *number of inner-loop updates*. Fig. 1 shows that the performance unstably degrades as either the learning rate or the number of inner-loop updates increases. This degradation coincides with a sharp rise in the fraction of samples whose importance ratios are clipped (middle rows in Fig. 1). These results support our hypothesis that uncontrolled out-of-range samples contribute to training instability.

**Entropy Collapse and Mode Concentration.** As the number of inner-loop updates increases for a query $q$, a growing fraction of samples enters the clipped regime, fundamentally altering the learning dynamics. In this regime, group-normalized query-wise advantages exhibit a winner-takes-all behavior: a small subset of responses consistently attains higher relative advantages and is repeatedly reinforced across inner-loop updates, leading to pronounced mode concentration. Meanwhile, importance-ratio clipping suppresses gradient updates for samples that substantially deviate from the reference policy, including low-probability but potentially informative responses, thereby restricting effective exploration. The interaction between advantage concentration and selective gradient suppression progressively reduces response diversity, resulting in a rapid decline in policy entropy, eventually leading to entropy collapse as

in the bottom rows in Fig. 1. This failure mode highlights a structural limitation of GRPO: the absence of a query-adaptive mechanism to balance exploration and exploitation beyond query-wise group averaging.

## 3. Method

We propose to reformulating the RL fine-tuning for LLMs as a *query-adaptive trust-region policy optimization problem*, to address the limitations identified in Sec. 6. Rather than relying on heuristic approximations to control policy updates, we propose to directly impose explicit KL-divergence constraints conditioned on each query (Sec. 3.1). Starting from this formulation, we derive a novel exact trust-region objective using Lagrangian duality. The resulting objective provides a principled optimization framework that explicitly controls how much each query contributes to the policy update based on its reward under the current policy. Interestingly, the resulting objective resembles a GSPO-style one with minor modifications. (Sec. 3.2) Lastly, we wrap up this section with intuitive explanation for the role of each component in the new objective (Sec. 3.3).

### 3.1. Prompt-wise Trust-Region Policy Optimization

**Problem Setting.** For each query, condition, or prompt $q$, our goal is to maximize the trajectory-level reward $R(o \mid q)$ while ensuring that the updated policy does not deviate too far from the previous policy. This leads to a prompt-conditioned trust-region optimization problem:

$$\sup_{\pi_\theta} \ \mathbb{E}_{o \sim \pi_\theta(\cdot|q)}[R(o \mid q)]$$
$$\text{s.t.} \quad \mathrm{KL}(\pi_\theta(\cdot \mid q) \| \pi_{\mathrm{old}}(\cdot \mid q)) \leq \delta, \tag{7}$$

where $\delta > 0$ controls the allowed policy deviation. This constraint is enforced independently for each prompt, providing explicit and continuous control over prompt-wise policy updates.

**Dual Formulation.** The optimization problem in Eq. (7) involves two constraints. The first is the trust-region constraint on the KL divergence, and the second is the normalization constraint requiring $\pi_\theta(\cdot \mid q)$ to be a valid probability distribution, *i.e.*, $\int \pi_\theta(o \mid q) \, \mathrm{d}o = 1$. Introducing dual variables $\lambda_q \geq 0$ and $\mu_q \in \mathbb{R}$ for these constraints, respectively, we obtain the following Lagrangian:

$$\mathcal{L}_q(\pi, \lambda_q, \mu_q) = \mathbb{E}_{o \sim \pi(\cdot|q)}[R(o \mid q)] + \lambda_q(\delta - \mathrm{KL}(\pi \| \pi_{\mathrm{old}}))$$
$$+ \mu_q \left( 1 - \int \pi(o \mid q) \, \mathrm{d}o \right). \tag{8}$$

For fixed $\lambda_q$ and $\mu_q$, taking the first variation of Eq. (8) with

respect to $\pi$ yields the optimal policy

$$\text{Eq. (7)} = \min_\pi \ \max_{\lambda_q \geq 0, \ \mu_q} \ \mathcal{L}_q(\pi, \lambda_q, \mu_q)$$
$$= \max_{\lambda_q \geq 0, \ \mu_q} \ \min_\pi \ \mathcal{L}_q(\pi, \lambda_q, \mu_q). \tag{9}$$

Given $\lambda_q$ and $\mu_q$, by taking first variation with respect to $\pi$, we obtain the following optimality condition for the inner optimization problem of Eq. (9):

$$\operatorname*{argmin}_\pi \ \mathcal{L}_q(\pi, \lambda_q, \mu_q) = \pi_{\mathrm{old}}(o \mid q) e^{\frac{R(o|q) - \mu_q}{\lambda_q} - 1}. \tag{10}$$

Now, by taking derivative with respect to dual variables $\mu_q$ and $\lambda_q$, we can derive the following property:

**Proposition 3.1.** *Let function* $f_q : \mathbb{R} \to \mathbb{R}$ *be defined as*

$$f_q(\lambda) = \lambda \left( \delta + \log \mathbb{E}_{o \sim \pi_{\mathrm{old}}(\cdot|q)} \exp\left( \frac{R(o \mid q)}{\lambda} \right) \right). \tag{11}$$

*Given that*

$$\lambda_q^\star := \min_{\lambda \geq 0} f_q(\lambda), \quad \mu_q^\star := f_q(\lambda_q^\star) - \lambda_q^\star(\delta + 1), \tag{12}$$

*the following holds:*

$$\pi^\star(o \mid q) = \pi_{\mathrm{old}}(o \mid q) \exp\left( \frac{R(o \mid q) - \mu_q^\star}{\lambda_q^\star} - 1 \right). \tag{13}$$

Detailed discussion and proof is provided in Appendix A.

**Remark 3.2.** The optimal dual variable $\lambda_q^\star$ acts as a query-specific regularization parameter that controls the sharpness of the policy update. When the reward distribution under query $q$ exhibits high variance, the log-partition term in Eq. (11) increases, leading to a larger optimal $\lambda_q^\star$. This results in a flatter exponential weighting and therefore a more conservative policy update. Conversely, prompts with more concentrated or stable reward distributions admit smaller $\lambda_q^\star$, allowing sharper updates. This automatic, prompt-dependent calibration of update magnitude is a direct consequence of solving the trust-region problem exactly, and cannot be achieved by global or heuristic KL penalties. We further discuss the role of each term in Sec. 3.3.

### 3.2. Objective Function and Algorithm

**Objective Function.** We now consider a parametric objective whose solution corresponds to the projection of the optimal policy $\pi^\star$.

**Theorem 3.3** (QUATRO Objective). *Given a prompt $q$ and* $\{o^{(i)}\}_{i=1}^N$, *consider the following objective:*

$$\mathcal{L}_q \left( \{o^{(i)}\}_{i=1}^N \right) = -\sum_{i=1}^N \left[ \frac{\pi_\theta(o^{(i)} \mid q)}{\pi_{\mathrm{old}}(o^{(i)} \mid q)} \right.$$
$$\left. \cdot \left( A_q^i - \log\left( \frac{\pi_{\bar\theta}(o^{(i)} \mid q)}{\pi_{\mathrm{old}}(o^{(i)} \mid q)} \right) \right) \right], \tag{14}$$

where $\pi_{\bar{\theta}}$ denotes a gradient-detached version of $\pi_\theta$ and

$$A_q^i = \frac{R(o^{(i)} \mid q) - \mu_q^\star}{\lambda_q^\star} - 1. \qquad (15)$$

Then, the optimal policy $\pi^\star$ in Eq. (13) can be written as

$$\pi^\star = \operatorname*{argmin}_{\pi_\theta} \mathbb{E}_{\{o^{(i)}\}_{i=1}^N \sim \pi_{\text{old}}(\cdot|q)} \mathcal{L}_q\left(\{o^{(i)}\}_{i=1}^N\right). \qquad (16)$$

**Remark 3.4** (Connection to GSPO)**.** Consider a simplified version of Eq. (14) in which the log-ratio term $\log\left(\pi_{\bar{\theta}}/\pi_{\text{old}}\right)$ is removed and the GSPO advantage is substituted for $A_q^{(i)}$. Under this modification, and with the addition of importance-ratio clipping, the resulting update reduces exactly to the GSPO objective. Therefore, the difference from GSPO is mainly that, our formulation retains the log-ratio term and with the advantage derived from the exact trust-region optimization instead of clipping. This term provides a principled mechanism for regulating policy updates, allowing the optimization to actively control deviations from the reference policy without relying on clipping. We further discuss the interpretation of this term in Sec. 3.3.

Many RL-based LLM post-training objectives regularize policy updates using a KL-divergence penalty to prevent excessive deviation from a pretrained model (Aminian et al., 2025; Ouyang et al., 2022; Rafailov et al., 2023). As our formulation similarly imposes a KL-divergence constraint, the resulting objective admits an optimal policy that can be interpreted as a *geometric interpolation* between the TR optimal policy $\pi^\star$ and the pretrained model $\pi_{\text{pre}}$.

**Theorem 3.5.** *Consider the following loss functional:*

$$\mathbb{E}_{\{o^{(i)}\}_{i=1}^N \sim \pi_{\text{old}}(\cdot|q)} \mathcal{L}_q\left(\{o^{(i)}\}_{i=1}^N\right) + \beta \mathrm{KL}(\pi_\theta \mid \pi_{pre}), \quad (17)$$

*where $\pi_{pre}$ is a pretrained LLM model. Then, the optimal policy $\pi_{kl}^\star$ obtained from minimization of Eq. (17) is*

$$\pi_{kl}^\star(\cdot \mid q) \propto \pi^\star(\cdot \mid q)^{\frac{1}{\beta+1}} \pi_{pre}(\cdot \mid q)^{\frac{\beta}{\beta+1}}. \qquad (18)$$

The proofs are provided in Appendix A.

**Approximating the Adjoint Variables.** We approximate the adjoint variables using Monte Carlo samples $\{o^{(i)}\}_{i=1}^N \sim \pi_{\text{old}}(\cdot \mid q)$ drawn from the reference (old) policy. Specifically, the dual objective $f_q$ in Eq. (11) is approximated by its empirical counterpart:

$$f_q(\lambda_q) \approx \lambda_q\left(\delta + \log \frac{1}{N}\sum_{i=1}^N \exp\left(\frac{R(o^{(i)} \mid q)}{\lambda_q}\right)\right). \quad (19)$$

We then obtain the optimal dual variable $\lambda_q^\star$ by minimizing Eq. (19). Computational overhead of this optimization is negligible, as it reduces to a scalar optimization.

**Algorithm 1** Adaptive Trust-Region Policy Optimization

**Require:** Initial pretrained policy $\pi_{\text{pre}}$, a reward model $R$, a prompt dataset $\mathcal{D}$, the number of rollouts per prompt $N$, and a TR radius $\delta$, the number of inner-loop update $K$.
1: Initialize $\pi_\theta \leftarrow \pi_{\text{pre}}$.
2: **for** iterations **do**
3:     Sample a batch of prompts $\mathcal{D}_b \subset \mathcal{D}$.
4:     Update old policy model $\pi_{\text{old}} \leftarrow \pi_\theta$
5:     Sample $\{o^{(i)}\}_{i=1}^N \sim \pi_{\text{old}}(\cdot \mid q)$ for $q \in \mathcal{D}_b$
6:     $R_i \leftarrow R(o^{(i)}|q)$ for each $i \in \{1,\dots,N\}$ and $q \in \mathcal{D}_b$.
7:     Optimize $(\lambda_q^\star, \mu_q^\star)$ following eq. (19) for every $q \in \mathcal{D}_b$.
8:     Compute advantages: $A_q^i = \frac{1}{\lambda_q^\star}(R_i - \mu_q^\star) - 1$.
9:     Update policy $\pi_\theta$ by maximizing eq. (17) for $K$ times.
10: **end for**
11: Return $\pi_\theta$

In principle, the estimation becomes more accurate as $N$ increases. However, as shown in Fig. 2 and App. C.1, we empirically find that a relatively small number of samples (e.g., $N = 8$) is sufficient to provide reliable learning signals.

**Algorithm.** As shown in Alg. 1, for each query $q$, we collect samples $\{o^{(i)}\}_{i=1}^N$ from $\pi_{\text{old}}$ and compute rewards $\{R_i\}_{i=1}^N$ for each sample, respectively. Then, we optimize dual variables to obtain $(\lambda_q^\star, \mu_q^\star)$ by using Eq. (19). Then, we compute advantage $A_q^i$ by Eq. (15), then minimize the loss Eq. (17) for the number of inner-loops $K$.

### 3.3. Interpretation and Key Observations

**Role of Log-Ratio Term.** We now examine the role of the term $\log \frac{\pi_\theta^i}{\pi_{\text{old}}^i}$ in Eq. (14), where $\pi^i := \pi(o^{(i)} \mid q)$. Considering the sign of the quantity $A_q^i - \log \frac{\pi_\theta(o^{(i)}|q)}{\pi_{\text{old}}(o^{(i)}|q)}$,

$$A_q^i - \log \frac{\pi_\theta^i}{\pi_{\text{old}}^i} \le 0 \iff \pi^\star(o^{(i)}|q) := e^{A_q^i}\pi_{\text{old}}^i \le \pi_\theta^i,$$

the loss decreases the probability assigned by the current policy $\pi_\theta$. That is, when $\pi_\theta$ assigns higher probability than the TR optimal policy $\pi^\star$, the update acts to suppress this overestimation. Conversely, when $A_q^i - \log \frac{\pi_\theta(o^{(i)}|q)}{\pi_{\text{old}}(o^{(i)}|q)} \ge 0$, the update increases $\pi_\theta(o^{(i)} \mid q)$, pushing the policy toward $\pi^\star$. In both cases, our loss provides a symmetric correction that steers the current policy toward the trust-region optimal policy without relying on heuristic clipping. Therefore, the log-ratio term serves as a stabilizer that adaptively regulates policy updates by counteracting over- and under-estimation relative to the trust-region optimum.

**Role of Dual Variables** $(\lambda^\star, \mu^\star)$**.** In GRPO and GSPO, the advantage $A^i$ is implicitly controlled through heuristic reward normalization, typically using group-wise statistics $A^i = \frac{R_i - \text{mean}(R_i)}{\text{std}(R_i)}$. In contrast, our method controls the advantage via the dual variables $(\lambda_q^\star, \mu_q^\star)$, yielding

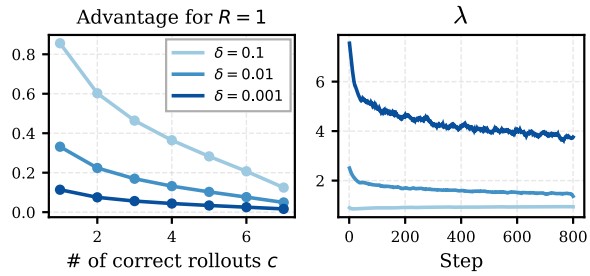

*Figure 2.* Prompt-wise update magnitude (left) and training-time evolution of the Lagrange multiplier $\lambda$ under different $\delta$ (right).

*Table 1.* **Average Pass@$k$ over six mathematical reasoning benchmarks** (MATH500, AMC23, AIME24, AIME25, Minerva-Math, and OlympiadBench). Our method shows increasing gains as the number of samples $k$ grows.

| Method | 1 | 2 | 4 | 8 | 16 | 32 | 64 | 128 | 256 |
|---|---|---|---|---|---|---|---|---|---|
| **Qwen2.5-Math-1.5B** | | | | | | | | | |
| GRPO | 31.66 | 37.28 | 42.15 | 46.29 | 49.79 | 53.14 | 56.62 | 60.42 | 64.36 |
| GPG | 32.03 | 37.71 | 42.63 | 46.77 | 50.20 | 53.41 | 56.85 | 60.57 | 64.25 |
| GMPO | 32.17 | 37.83 | 42.78 | 47.01 | 50.49 | 53.56 | 56.72 | 60.10 | 63.85 |
| GSPO | 32.15 | 37.56 | 42.36 | 46.53 | 50.13 | 53.24 | 55.92 | 58.29 | 60.85 |
| DAPO | 31.85 | 37.05 | 41.56 | 45.49 | 48.92 | 51.96 | 54.90 | 58.45 | 63.02 |
| CISPO | 31.60 | 36.91 | 41.64 | 45.81 | 49.57 | 53.08 | 56.64 | 60.46 | 64.32 |
| Ours $_{\delta=0.1}$ | **32.63** | **38.26** | **42.90** | 46.75 | 50.23 | 53.63 | 57.12 | 60.61 | 63.65 |
| Ours $_{\delta=0.01}$ | 31.02 | 37.12 | 42.45 | **47.04** | 51.02 | 54.67 | 58.11 | 61.75 | 65.79 |
| Ours $_{\delta=0.001}$ | 29.73 | 36.18 | 41.91 | 46.96 | **51.52** | **55.82** | **59.96** | **64.10** | **68.68** |
| **Qwen2.5-Math-7B** | | | | | | | | | |
| GSPO | 40.05 | 45.34 | 49.53 | 53.08 | 56.24 | 59.02 | 61.45 | 63.65 | 65.70 |
| Ours $_{\delta=0.01}$ | **40.08** | **46.14** | **51.00** | **55.11** | **58.77** | 62.18 | 65.55 | 68.84 | 71.72 |
| Ours $_{\delta=0.001}$ | 37.82 | 44.38 | 49.67 | 54.29 | 58.59 | **62.58** | **66.38** | **70.02** | **73.41** |

$A_q^i = \frac{R_i - (\lambda_q^\star + \mu_q^\star)}{\lambda_q^\star}$ as shown in Eq. (15). From this perspective, $\lambda_q^\star$ plays a role analogous to the standard deviation term in GRPO or GSPO, but its scale is jointly determined by the trust-region radius $\delta$ and the reward distribution $\{R_i\}$ for each query $q$, rather than by heuristic normalization. Finally, the trust-region bound $\delta$ directly regulates the update magnitude; a larger $\delta$ yields a smaller $\lambda_q^\star$, amplifying the advantage, allowing larger policy updates. This aligns with the intuition that a looser trust-region constraint permits more aggressive updates, while a tighter bound enforces more conservative policy changes.

Fig. 2 shows that the trust-region budget $\delta$ explicitly controls update magnitude through the dual variable $\lambda$. Tighter trust regions induce larger values of $\lambda$, resulting in uniformly attenuated prompt-wise advantages. As a consequence, update magnitude decreases as the number of correct rollouts increases, with smaller $\delta$ consistently yielding more conservative policy updates.

# 4. Experiments

## 4.1. Experimental Setup

**Task and Datasets.** We evaluate our method on the mathematical reasoning task. Specifically, we fine-tune pre-

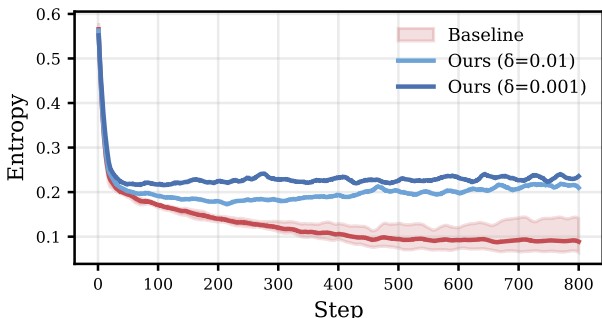

*Figure 3.* **Entropy dynamics comparison across training steps**. The baseline exhibits a steady entropy collapse, while our method maintains it at a controlled and stable level throughout training.

*Table 2.* **Final policy entropy after training.** Our method maintains higher entropy than prior approaches.

| | Baselines | | | Ours | | |
|---|---|---|---|---|---|---|
| Method | GSPO | DAPO | CISPO | $\delta=0.1$ | $\delta=0.01$ | $\delta=0.001$ |
| Entropy | 0.05 | 0.07 | 0.05 | 0.14 | 0.20 | **0.23** |

trained Qwen2.5-Math-1.5B and Qwen2.5-Math-7B (Yang et al., 2024) models on the full MATH (Hendrycks et al., 2021; Lightman et al., 2023) dataset, consisting of 7,500 problems, and evaluate on six mathematical benchmarks: MATH500 (Hendrycks et al., 2021), AMC23 (MAA, 2023), AIME24/25 (MAA, 2025), MINERVA MATH (Lewkowycz et al., 2022), and OLYMPIAD-BENCH (Huang et al., 2024).

**Baselines.** We compare against GRPO (Shao et al., 2024), GPG (Chu et al., 2026), GSPO (Zheng et al., 2025a), GMPO (Zhao et al., 2026), DAPO (Yu et al., 2025) and CISPO (Chen et al., 2025a) as baselines.

**Evaluation Metrics.** We measure the standard unbiased Pass@$k$ metric (Chen, 2021; Peng et al., 2025), indicating the hit ratio of problems with $k$ chances to answer, using $k \in \{1, 2, 4, 8, 16, 32, 64, 128, 256\}$. Formally, Pass@$k$ is defined as $\mathbb{E}\left[1 - \binom{n-c}{k}/\binom{n}{k}\right]$, where $c$ denotes the number of correct completions out of $n$ sampled responses. Following prior work, we use $n = 256$ for all datasets.

Although Pass@$k$ has been used as a standard metric for evaluating stochastic mathematical reasoning models, it does not distinguish between discovering multiple distinct correct solutions *v.s.* repeatedly generating the same correct solution, as it just measures whether at least one correct answer has been found among the $k$ trials. To address this limitation, we introduce a new metric named **Unique Correct Count (UCC)@$k$**, which measures the number of *unique correct solutions* observed among the $k$ samples. To determine if two answers are identical, we group semantically identical correct answers into clusters, since a naive text

*Table 3.* UCC@$k$ across six mathematical reasoning benchmarks (MATH500, AMC23, AIME24, AIME25, MinervaMath, and Olympiad-Bench). Higher values indicate a larger number of unique correct solutions.

| Dataset | 32 | | | | | 64 | | | | | 128 | | | | | 256 | | | | |
|---|---|---|---|---|---|---|---|---|---|---|---|---|---|---|---|---|---|---|---|---|
| | GRPO | GPG | GMPO | GSPO | Ours | GRPO | GPG | GMPO | GSPO | Ours | GRPO | GPG | GMPO | GSPO | Ours | GRPO | GPG | GMPO | GSPO | Ours |
| MATH500 | 4.04 | 10.76 | 5.30 | 5.20 | **13.17** | 6.29 | 20.37 | 9.10 | 8.86 | **25.65** | 10.00 | 38.66 | 15.89 | 15.30 | **49.97** | 16.17 | 73.25 | 27.98 | 26.66 | **97.20** |
| AMC23 | 5.40 | 11.83 | 6.95 | 7.27 | **13.23** | 9.20 | 22.71 | 12.48 | 13.14 | **25.96** | 15.76 | 43.71 | 22.86 | 24.02 | **51.09** | 27.02 | 84.05 | 42.38 | 44.02 | **100.62** |
| AIME24 | 1.88 | 3.68 | 2.82 | 2.48 | **3.77** | 3.22 | 7.16 | 5.12 | 4.53 | **7.47** | 5.53 | 13.92 | 9.31 | 8.31 | **14.80** | 9.60 | 27.03 | 16.87 | 15.13 | **29.33** |
| AIME25 | 0.83 | 1.57 | 1.37 | 1.26 | **1.36** | 1.42 | 3.04 | 2.53 | 2.32 | **2.67** | 2.44 | 5.93 | 4.70 | 4.32 | **5.27** | 4.20 | 11.57 | 8.73 | 8.07 | **10.43** |
| MinervaMath | 2.14 | 2.99 | 2.30 | 2.38 | **3.24** | 3.87 | 5.70 | 4.22 | 4.35 | **6.30** | 7.03 | 10.92 | 7.80 | 8.01 | **12.25** | 12.80 | 20.87 | 14.44 | 14.80 | **23.79** |
| OlympiadBench | 2.99 | 5.00 | 3.35 | 3.52 | **15.80** | 5.28 | 9.51 | 6.09 | 6.43 | **11.32** | 9.43 | 18.14 | 11.15 | 11.84 | **22.08** | 16.93 | 34.61 | 20.48 | 21.89 | **42.97** |

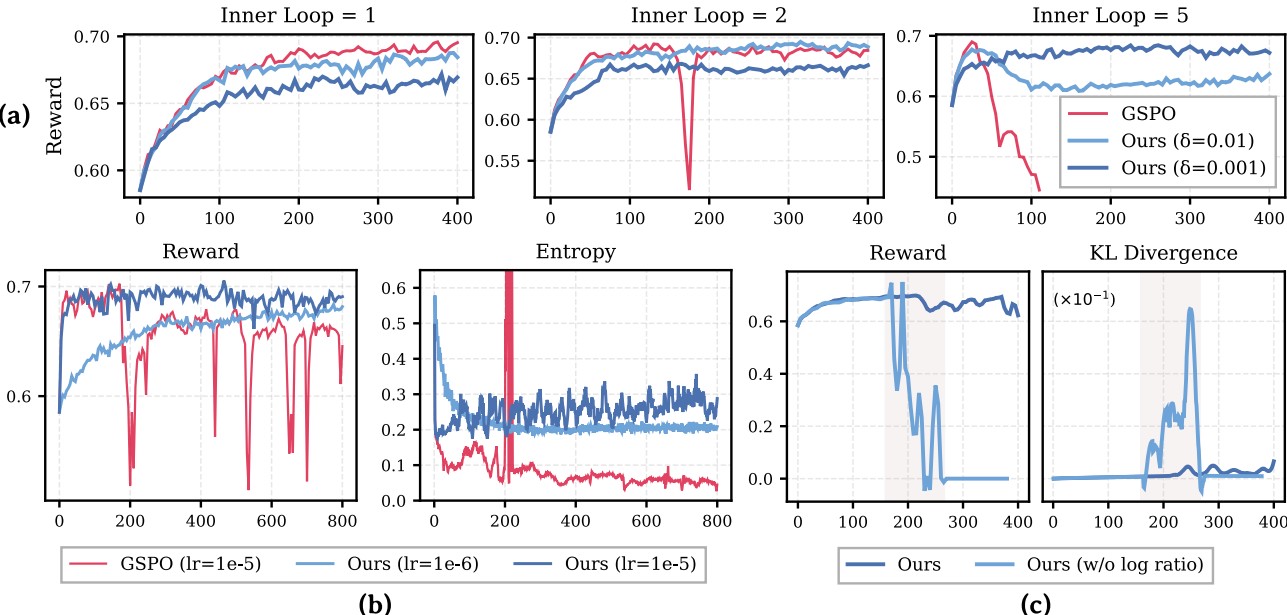

*Figure 4.* **Robustness to policy staleness and aggressive updates.** Both increased offline inner-loop updates (a) and larger learning rates (b) amplify policy mismatch, leading GSPO to unstable rewards and entropy collapse. In contrast, our method remains stable across these settings, enabled by explicit trust-region control and the log-ratio stabilizer (c).

match would not catch nearly identical but slightly different ones. We adopt a commonly-used text similarity metrics, *e.g.*, TF-IDF cosine similarity, where each cluster corresponds to a distinct reasoning solution. (See more similarity metrics in Appendix C.5.) Formally, UCC@$k$ is defined as

$$\text{UCC@}k = \sum_{j=1}^{m} \left( 1 - \frac{\binom{n-s_j}{k}}{\binom{n}{k}} \right), \quad (20)$$

where $n$ is the total number of generated samples, $s_j$ is the number of samples belonging to the $j$-th correct cluster. The term inside the summation corresponds to the probability that at least one sample from the cluster $j$ is included in the given $k$ samples. In this sense, UCC@$k$ complements Pass@$k$ by capturing the *diversity of correct solutions*, providing a more faithful indicator of reasoning robustness.

**Implementation Details.** For each query, we generate 8 rollouts per prompt, with maximum response lengths of 1,024 and 3,072 tokens for the 1.5B and 7B models, respectively. We set the batch size to 256 and learning rate to $10^{-6}$. We use the AdamW optimizer. We use top-$p$ 1.0 at training,

and top-$p$ 0.7 at evaluation, set the temperature to 1.0. See Tab. I in Appendix B for more hyperparameter settings.

### 4.2. Results and Analysis

**Overall Performance.** We first evaluate overall reasoning accuracy using the standard Pass@$k$ metric. Tab. 1 reports the Pass@$k$ scores averaged over all benchmarks, following Peng et al. (2025). (See Tab. II in Appendix C.3 for dataset-wise results.) Across all datasets and model scales (1.5B and 7B), our method consistently outperforms all other baselines, with the performance gap becoming more pronounced with larger $k$. Specifically, a larger $\delta$ results in a smaller $\lambda$, which sharply emphasizes reward differences and reinforces dominant correct solution patterns, leading to higher accuracy. Conversely, a smaller $\delta$ tends to yield a larger $\lambda$ (see Appendix C.1), which smooths reward differences and maintains multiple valid reasoning trajectories, increasing solution diversity. By adjusting $\delta$, we can explicitly choose the desired policy behavior depending on whether accuracy (*exploitation*) or diversity (*exploration*) is prioritized. Additionally, we included the flip-rate analysis,

Appendix C.7, measuring the fraction of queries that transition from unsolved under the base policy to solved after fine-tuning, and analysis of unique correct answer ratios under varying decoding temperatures.

**Entropy Stability and Optimization Behavior.** Beyond the accuracy, we measure token-level policy entropy during generation to track diversity throughout training (see Appendix C.2 for formal definition). As shown in Fig. 3, the baseline such as GSPO exhibits rapid entropy collapse, converging to extremely low entropy early at training. In contrast, our method maintains stable entropy throughout the training, without suffering from premature distribution sharpening. This behavior aligns with prior reports that the entropy collapse is closely linked to reduced exploration and degraded reasoning robustness (Chen et al., 2025b; He et al., 2025). Recent methods such as DAPO and CISPO also aim to mitigate entropy collapse and achieve improved entropy compared to earlier baselines. However, as shown in Tab. 2, their final entropy remains significantly lower than that of our method. Our approach not only stabilizes entropy throughout training, but also maintains a substantially higher entropy at convergence, enabling more effective exploration without sacrificing task performance. By explicitly regulating the update through a principled trust-region formulation, our method achieves stable optimization without relying on heuristic clipping, leading to a more reliable training.

**Diversity in UCC@$k$.** Tab. 3 reports UCC@$k$ with $k \in \{32, 64, 128, 256\}$ (results with smaller $k$ available in Appendix C.4) across the six benchmarks. Across all datasets and sampling budgets, our method consistently achieves significantly higher UCC@$k$ than the baselines. A higher UCC@$k$ implies that the model generates diverse correct reasoning paths (*i.e.*, a larger number of *unique correct solutions*), rather than repeated variations of a single solution.

We further provide a complementary analysis of output diversity by measuring multiple pairwise similarity metrics among correct answers (*e.g.*, ROUGE-L and token-level Jaccard similarity), with qualitative examples in Appendix C.6.

**Generalization to Code Generation.** We evaluate our method on code generation tasks to assess its generalization beyond mathematical reasoning. We first fine-tune Qwen2.5-Coder-1.5B on the APPS dataset and evaluate on HumanEval and HumanEval+. As shown in Table X, our method consistently outperforms the GSPO baseline across all Pass@$k$ metrics, with more pronounced gains at larger $k$. Additionally, we conduct zero-shot evaluation on MBPP and MBPP+. Our method again achieves consistent improvements over GSPO, with larger gains at Pass@16. These results suggest that the advantages of our method are not limited to mathematical reasoning, but extend to code generation tasks.

*Table 4.* **Code generation performance (Pass@$k$).** Our method consistently outperforms GSPO on HumanEval(+) and MBPP(+), with more pronounced gains at larger $k$.

| HumanEval | | | | | |
|---|---|---|---|---|---|
| Method | 1 | 4 | 8 | 16 | 32 | 64 |
| GSPO | 2.42 | 8.22 | 13.93 | 21.83 | 31.18 | 40.85 |
| Ours | **2.94** | **10.07** | **17.23** | **27.19** | **38.67** | **50.00** |

| HumanEval+ | | | | | |
|---|---|---|---|---|---|
| Method | 1 | 4 | 8 | 16 | 32 | 64 |
| GSPO | 1.98 | 6.82 | 11.72 | 18.72 | 27.37 | 37.20 |
| Ours | **2.35** | **8.06** | **13.85** | **22.08** | **31.85** | **41.46** |

| MBPP / MBPP+ (zero-shot) | | | | | |
|---|---|---|---|---|---|
| | MBPP | | | MBPP+ | | |
| Method | 1 | 8 | 16 | 1 | 8 | 16 |
| GSPO | 5.41 | 29.85 | 44.71 | 4.81 | 26.98 | 40.62 |
| Ours | **6.19** | **33.77** | **49.47** | **5.39** | **29.52** | **43.39** |

### 4.3. Stability Analysis

Motivated by failure case in Sec. 6 and Fig. 1, we evaluate the robustness of our method under increasingly aggressive optimization regimes, including greater policy staleness, and larger update steps. Our experiments validate that our method remains robust as the mismatch between the sampling policy and the updated policy grows.

**Stability under Policy Staleness.** Increasing the number of inner-loop updates is a common strategy to improve sample efficiency by reusing collected rollouts. However, this inevitably increases policy staleness, which often leads to instability in clipping-based methods. As shown in Fig. 4 (a), our method remains stable as the number of inner-loop updates increases, whereas baseline methods exhibit reward degradation or collapse.

**Stability under Large Learning Rates.** We next examine robustness under aggressive step sizes by increasing the learning rate, a common approach to accelerate convergence and reduce training time. Fig. 4(b) shows that our method tolerates significantly larger learning rates without instability, while baseline methods quickly diverge. This robustness allows for faster convergence in practice, demonstrating that our trust-region formulation supports both stability and efficient optimization under aggressive learning settings.

**Role of the Log-Ratio Penalty.** As discussed in Sec. 3.3, the log-ratio term is theoretically motivated to mitigate reward over-estimation. To rigorously evaluate the stabilizing property of the log-ratio penalty, we conduct the ablation in a high variance regime ($\delta = 0.1$, two inner-loop updates), chosen to induce the instabilities during aggressive policy optimization. Fig. 4(c) empirically confirms this effect: removing the log-ratio term leads to reward collapse accompanied by abrupt spikes in KL divergence. In contrast, incorporating the log-ratio penalty yields stable reward improvement with well-controlled KL dynamics, validating its stabilizing role in practice.

## 4.4. Computational Cost

The additional computational cost of our method is minimal, as it only involves solving a per-query scalar dual variable $\lambda_q$ using already-sampled trajectories and reward statistics, without requiring additional rollouts or auxiliary model training. In practice, rollout generation and log-probability evaluation dominate the cost, accounting for the majority of the total computation (approximately 60–90%). In contrast, the remaining components, including advantage computation and dual optimization, contribute only a small fraction. Overall, the total training time is nearly identical to the baseline, indicating negligible overhead.

## 5. Related Work

Several recent works aim to improve the stability of GRPO-style critic-free training for reasoning LLMs (Shao et al., 2024; Liu et al., 2025; Chu et al., 2026; Zheng et al., 2025a; Zhao et al., 2026). Prior analyses show that GRPO-style methods are prone to instability and distribution sharpening, particularly under stale samples or aggressive updates. (Liu et al., 2025; Mroueh et al., 2026). To mitigate this, existing approaches regulate exploration dynamics through heuristic mechanisms, including modified importance-weight clipping, gradient-preserving updates, and second-moment–based constraints (Yu et al., 2025; Su et al., 2025; 2026; Chen et al., 2025a; Deng et al., 2025; Yang et al., 2025b; Xi et al., 2026; Zheng et al., 2025b). Other works incorporate uncertainty- or entropy-aware reweighting into the advantage signal, leveraging semantic uncertainty estimation, entropy–gradient coupling, or entropy-modulated objectives to temper aggressive updates (Chen et al., 2025b; Wang et al., 2025b; Cheng et al., 2026; Zhang et al., 2025; Wang et al., 2025a). Other works introduce entropy regularization via covariance-based penalties or activation-level constraints to stabilize gradient flow without directly modifying the policy objective (Cui et al., 2025; Kang et al., 2026). Collectively, these methods stabilize RL training by heuristically shaping entropy dynamics.

A related line of work addresses imbalance in learning signals across queries with heterogeneous difficulty and reward sparsity. These methods adjust advantage baselines, loss weighting, or sampling strategies based on reward statistics, confidence, quantiles, or uncertainty to emphasize difficult or informative queries (Wu et al., 2026; Ren et al., 2026; He et al., 2025; Zhang et al., 2026a; Chen et al., 2026; Bamba et al., 2025; Chen et al., 2025c; Zhang et al., 2026b). To improve robustness under partial off-policy sampling, several approaches adopt heuristic trust-region approximations (Zheng et al., 2025b; Deng et al., 2025; Roux et al., 2025; Fu et al., 2025; Arnal et al., 2025). In contrast, KL-regularized policy gradient methods provide a more principled foundation for controlling policy deviation (Zhang et al., 2026c). A closely related objective is the *proximal exact policy gradient* loss of Choi et al. (2026), which uses a similar proximal formulation but does not explicitly incorporate trust-region bound computation.

However, existing trust-region and KL-regularized methods rely on global hyperparameters, lacking query-conditioned control for instance-level optimization. We instead formulate RLVR as a query-wise constrained problem and derive a closed-form update with query-specific dual variables, enabling principled trust-region control.

## 6. Limitation

Our method introduces the KL constraint parameter $\delta$ to control policy updates in a principled manner. While $\delta$ provides an explicit handle over the exploration–exploitation trade-off, its selection currently remains a hyperparameter choice. Designing adaptive strategies for $\delta$ selection is an interesting direction for future work.

Moreover, the effectiveness of our approach may depend on task-specific factors such as reward distribution and response diversity. In particular, it may be beneficial in settings with continuous or fine-grained rewards. Further empirical validation across diverse reward settings is left for future work.

## 7. Conclusion

QUATRO is a query-adaptive trust-region policy optimization framework that enforces per-query KL constraints via dual updates, stabilizing training, preserving policy entropy, and eliminating the need for importance-ratio clipping. Our experiments show that QUATRO outperforms GRPO-style baselines in Pass@$k$, with larger gains at higher $k$, maintaining stable entropy under policy staleness and aggressive learning rates. We discover that these improvements arise from increased diversity among correct solutions rather than repeated sampling of identical answers.

## Acknowledgments

This work was also supported by Samsung Electronics, Youlchon Foundation, National Research Foundation of Korea (NRF) grants (RS-2021-NR05515, RS-2024-00336576, RS-2023-0022663, RS-2025-25399604, RS-2024-00342044, RS-2025-16063688, RS-2025-02215813, RS-2026-25491306), the Institute for Information & Communication Technology Planning & Evaluation (IITP) grants (RS-2022-II220264, RS-2024-00353131), and the "Advanced GPU Utilization Support Program" (No. 02-26-01-0303) funded by the Ministry of Science and ICT of the Korean government.

## Impact Statement

This paper aims to advance the field of machine learning by improving the stability and robustness of reinforcement learning methods for training large language models. By mitigating known failure modes such as mode collapse and training instability, our work may contribute to more reliable deployment of reasoning-capable models. We do not anticipate specific negative ethical or societal impacts beyond those already studied in the context of large language models.

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

# A. Proofs

In this section, we denote the trajectory as $\tau$, and trajectory $\tau$ sampled from the sampling (old) policy $\pi_{\text{old}}$ as $\tau \sim \pi_{\text{old}}$. For convenience, we omit the query or condition $q$ when there is no confusion.

## A.1. Derivation of the Exact Query-wise Trust-Region Policy Update

**Constrained Optimization Problem**   We aim to optimize the expected reward under a new policy $\pi_\theta(\tau \mid q)$ for a given prompt $q$, while ensuring that it does not deviate too far from a old policy $\pi_{\text{old}}(\tau \mid q)$. Formally, we consider the following constrained optimization problem (Schulman et al., 2015; Guo et al., 2026; Tomar et al., 2020; Blessing et al., 2025; Abdolmaleki et al., 2015; Otto et al., 2021; Peters et al., 2010):

$$\sup_{\pi_\theta} \mathbb{E}_{\tau \sim \pi_\theta(\cdot \mid q)}[R(\tau \mid q)] \quad \text{s.t.} \quad \text{KL}(\pi_\theta(\cdot \mid q) \,\|\, \pi_{\text{old}}(\cdot \mid q)) \leq \delta. \tag{21}$$

Here, $\tau$ denotes a trajectory sampled from $\pi_\theta(\cdot \mid q)$, and $R(\tau \mid q)$ is the reward associated with trajectory $\tau$ conditioned on the prompt $q$. To handle the constraint, we introduce a Lagrange multiplier $\lambda \geq 0$ for the KL constraint and a multiplier $\mu$ for the normalization constraint (i.e., $\int \pi_\theta(\tau \mid q) \, d\tau = 1$). This leads to the following dual formulation:

$$\sup_{\pi_\theta} \inf_{\lambda \geq 0, \mu} \mathcal{L}(\pi_\theta, \lambda, \mu), \quad \text{where}$$

$$\mathcal{L}(\pi_\theta, \lambda, \mu) = \mathbb{E}_{\tau \sim \pi_\theta(\cdot \mid q)} \left[ R(\tau \mid q) \right] + \lambda \left( \delta - \text{KL}(\pi_\theta(\cdot \mid q) \,\|\, \pi_{\text{old}}(\cdot \mid q)) \right) + \mu \left( 1 - \int \pi_\theta(\tau \mid q) \, d\tau \right). \tag{22}$$

This formulation makes explicit the dependence on the input prompt $q$, and expresses the optimization as a saddle-point problem over the policy $\pi_\theta$ and dual variable $\lambda$.

**Derivation of Optimal Policy via Lagrangian Duality** By leveraging dual formulation, we could describe the optimality condition, through dual variables, which can be derived as follows

**Step 1: Strong Duality** The convexity of the constraint (e.g., KL-divergence is convex w.r.t. $\pi_\theta$) and the linearity of the objective with respect to dual variables $(\lambda, \mu)$ holds. Thus, the following strong duality (Auslender & Teboulle, 2000; Boyd & Vandenberghe, 2004; Choi et al., 2023; 2024) holds:

$$\sup_{\pi_\theta} \inf_{\lambda \geq 0, \mu} \mathcal{L}(\pi_\theta, \lambda, \mu) = \inf_{\lambda \geq 0, \mu} \sup_{\pi_\theta} \mathcal{L}(\pi_\theta, \lambda, \mu). \tag{23}$$

Given dual variables $(\lambda, \mu)$, we first solve inner-loop optimization problem to describe the optimal policy $\pi^\star$.

**Step 2: Description of Optimality of $\pi_\theta(\tau \mid q)$ via Dual Variables**

Recall the KL divergence between two distributions:

$$\text{KL}(\pi \,\|\, \pi_{\text{old}}) = \int \pi(\tau \mid q) \log \frac{\pi(\tau \mid q)}{\pi_{\text{old}}(\tau \mid q)} \, d\tau. \tag{24}$$

Substituting this into the Lagrangian:

$$\mathcal{L}(\pi, \lambda, \mu) = \int \pi(\tau \mid q) R(\tau \mid q) \, d\tau - \lambda \int \pi(\tau \mid q) \log \frac{\pi(\tau \mid q)}{\pi_{\text{old}}(\tau \mid q)} \, d\tau + \lambda \delta + \mu \left( 1 - \int \pi(\tau \mid q) \, d\tau \right).$$

To find the optimal policy $\pi^*(\tau \mid q)$, we take the functional derivative (Evans, 2022) of $\mathcal{L}$ with respect to $\pi(\tau \mid q)$:

$$\frac{\delta \mathcal{L}}{\delta \pi(\tau \mid q)} = R(\tau \mid q) - \lambda \left( \log \frac{\pi(\tau \mid q)}{\pi_{\text{old}}(\tau \mid q)} + 1 \right) - \mu = 0. \tag{25}$$

From the previous derivation, we obtained:

$$\log \frac{\pi^*(\tau \mid q)}{\pi_{\text{old}}(\tau \mid q)} = \frac{R(\tau \mid q) - \mu}{\lambda} - 1 \Rightarrow \quad \pi^*(\tau \mid q) = \pi_{\text{old}}(\tau \mid q) \cdot \exp \left( \frac{R(\tau \mid q) - \mu}{\lambda} - 1 \right). \tag{26}$$

This expression reveals that the **policy ratio** – i.e., how much more (or less) likely a trajectory $\tau$ is under the optimized policy $\pi$ compared to the old policy $\pi_{\text{old}}$ – is directly governed by the trajectory-level reward $R(\tau \mid q)$.

## A.2. Deriving Optimal Dual Variable

**Optimality Condition for $\mu$**

We begin by plugging the optimality condition by using the normalization constraint:

$$
\begin{aligned}
1 = \int \pi^{\star}(\tau \mid q)\, d\tau &= \int \pi_{\text{old}}(\tau \mid q) \cdot \exp\left(\frac{R(\tau \mid q) - \mu}{\lambda} - 1\right) d\tau \\
&= \exp\left(-\frac{\mu}{\lambda} - 1\right) \cdot \int \pi_{\text{old}}(\tau \mid q) \cdot \exp\left(\frac{R(\tau \mid q)}{\lambda}\right) d\tau.
\end{aligned}
\tag{27}
$$

By organizing it, we obtain

$$
\exp\left(-\frac{\mu}{\lambda} - 1\right) = \left(\int \pi_{\text{old}}(\tau \mid q) \cdot \exp\left(\frac{R(\tau \mid q)}{\lambda}\right) d\tau\right)^{-1}.
\tag{28}
$$

Taking logarithms and rearranging, we obtain

$$
\begin{aligned}
-\frac{\mu}{\lambda} - 1 &= -\log\left(\int \pi_{\text{old}}(\tau \mid q) \cdot \exp\left(\frac{R(\tau \mid q)}{\lambda}\right) d\tau\right) \\
\implies \mu &= \lambda \cdot \left(\log\left(\int \pi_{\text{old}}(\tau \mid q) \cdot \exp\left(\frac{R(\tau \mid q)}{\lambda}\right) d\tau\right) - 1\right)
\end{aligned}
\tag{29}
$$

Alternatively, using the measure notation $d\pi_{\text{old}}$, we may write:

$$
\mu = \lambda \cdot \left(\log\left(\int \exp\left(\frac{R(\tau \mid q)}{\lambda}\right) d\pi_{\text{old}}(\tau \mid q)\right) - 1\right)
\tag{30}
$$

**Optimality Condition for $\lambda$**

We start from the dual formulation

$$
\inf_{\lambda \geq 0, \mu} \sup_{\pi} \left[\mathbb{E}_{\tau \sim \pi(\cdot \mid q)}[R(\tau \mid q)] + \lambda(\delta - \text{KL}(\pi(\cdot \mid q) | \pi_{\text{old}}(\cdot \mid q))) + \mu\left(1 - \int \pi(\tau \mid q) d\tau\right)\right].
\tag{31}
$$

Plugging the optimality condition eq. (26), we obtain

$$
\begin{aligned}
&\inf_{\lambda \geq 0} \left[\mathbb{E}_{\pi^{\star}}[R(\tau \mid q)] + \lambda\left(\delta - \left(\frac{1}{\lambda}\mathbb{E}_{\pi^{\star}}[R(\tau \mid q)] - \frac{\mu}{\lambda} - 1\right)\right)\right] \\
&= \inf_{\lambda \geq 0} \left[\mathbb{E}_{\pi^{\star}}[R(\tau \mid q)] + \lambda\delta - \mathbb{E}_{\pi^{\star}}[R(\tau \mid q)] + \mu + \lambda\right] \\
&= \inf_{\lambda \geq 0} \left[\lambda(\delta + 1) + \mu\right]
\end{aligned}
\tag{32}
$$

Substituting eq. (30), we obtain optimization problem for $\lambda$.

$$
\begin{aligned}
&\inf_{\lambda \geq 0} \left[\lambda(\delta + 1) + \lambda\left(\log\left(\int \exp\left(\frac{R(\tau \mid q)}{\lambda}\right) d\pi_{\text{old}}(\tau \mid q)\right) - 1\right)\right] \\
&= \inf_{\lambda \geq 0} \left[\lambda\left(\delta + \log\int \exp\left(\frac{R(\tau \mid q)}{\lambda}\right) d\pi_{\text{old}}(\tau \mid q)\right)\right] \\
&\approx \inf_{\lambda \geq 0} \left[\lambda\left(\delta + \log\frac{1}{N}\sum_{i=1}^{N}\exp\left(\frac{R(\tau_i \mid q)}{\lambda}\right)\right)\right] = \inf_{\lambda} f(\lambda).
\end{aligned}
\tag{33}
$$

Thus, we first obtain $\lambda^{\star} = \arg\inf_{\lambda \geq 0} f(\lambda)$, and obtain $\mu^{\star}$ through eq. (19). Then, plugging the dual variables $(\lambda^{\star}, \mu^{\star})$ into eq. (26), we could obtain an expression of optimal $\pi^{\star}$ as eq. (13).

## A.3. Loss Derivation

We redefine the training objective as the maximization of the negative KL divergence between the current policy $\pi_\theta$ and a target policy $\pi^*$. The optimization objective is

$$
\begin{aligned}
-D_{\mathrm{KL}}(\pi_\theta \| \pi^*) &= \int_{\mathcal{T}} \frac{\pi_\theta(\tau \mid q)}{\pi_{\mathrm{old}}(\tau \mid q)} \log \frac{\pi^*}{\pi_\theta}(\tau) \, d\pi_{\mathrm{old}}(\tau) \\
&= \int_{\mathcal{T}} \frac{\pi_\theta(\tau \mid q)}{\pi_{\mathrm{old}}(\tau \mid q)} \left[ \underbrace{\log \frac{\pi^*}{\pi_{\mathrm{old}}}(\tau)}_{\text{derived at eq(26)}} - \log \frac{\pi_\theta(\tau \mid q)}{\pi_{\mathrm{old}}(\tau \mid q)} \right] d\pi_{\mathrm{old}}(\tau)
\end{aligned} \tag{34}
$$

**Deriving our objective**

$$
J(\theta) = \frac{1}{N} \sum_{i=1}^{N} \left\{ \left( \frac{R_q^{(i)} - \mu_q}{\lambda_q} - 1 \right) \frac{\pi_\theta(\tau^{(i)} \mid q)}{\pi_{\mathrm{old}}(\tau^{(i)} \mid q)} - \frac{\pi_\theta(\tau^{(i)} \mid q)}{\pi_{\mathrm{old}}(\tau^{(i)} \mid q)} \log \frac{\pi_\theta(\tau^{(i)} \mid q)}{\pi_{\mathrm{old}}(\tau^{(i)} \mid q)} \right\}
$$

$$
\mathcal{L}(\theta) = -J(\theta) = -\frac{1}{N} \sum_{i=1}^{N} \frac{\pi_\theta(\tau^{(i)} \mid q)}{\pi_{\mathrm{old}}(\tau^{(i)} \mid q)} \left[ \frac{R_q^{(i)} - \mu_q}{\lambda_q} - 1 - \log \frac{\pi_\theta(\tau^{(i)} \mid q)}{\pi_{\mathrm{old}}(\tau^{(i)} \mid q)} \right]. \tag{35}
$$

Finally, we can redefine the advantage term in our dual formulation, which is consistent with GRPO's advantage estimator:

$$
A_q^i = \frac{R_q^{(i)} - \mu_q}{\lambda_q} - 1. \tag{36}
$$

As discussed in Sec. 3.3, replacing the log-ratio term $\log \frac{\pi_\theta(\tau^{(i)}|q)}{\pi_{\mathrm{old}}(\tau^{(i)}|q)}$ with its detached counterpart in $\mathcal{L}$ yields the same optimality condition.

## A.4. Proof of Thm. 3.5

By simply taking

$$
\mathcal{L}(\pi_\theta) = \mathbb{E}_{\tau \sim \pi_{\mathrm{old}}(\cdot | q)} \left[ \left( \frac{R(\tau) - \mu^\star}{\lambda^\star} - 1 - \log \frac{\pi_{\bar\theta}}{\pi_{\mathrm{old}}}(\tau) \right) \frac{\pi_\theta}{\pi_{\mathrm{old}}}(\tau) \right] - \beta \mathrm{KL}(\pi_\theta \, \| \, \pi_{\mathrm{pre}}). \tag{37}
$$

By taking first variation, we can obtain the optimality condition:

$$
C = \frac{\delta \mathcal{L}}{\delta \pi}(\pi^\star; \tau) = \frac{R(\tau) - \mu^\star}{\lambda^\star} - 1 - \log \frac{\pi^\star}{\pi_{\mathrm{old}}}(\tau) - \beta \log \frac{\pi^\star}{\pi_{\mathrm{pre}}}(\tau), \tag{38}
$$

hence,

$$
\pi^\star(\tau) \propto \left( e^{\frac{R(\tau) - \mu^\star}{\lambda^\star} - 1} \pi_{\mathrm{old}}(\tau) \right)^{\frac{1}{\beta+1}} \pi_{\mathrm{pre}}(\tau)^{\frac{\beta}{\beta+1}}. \tag{39}
$$

Thus, satisfying eq. (17).

# B. Implementation Details

**Framework.** All experiments are conducted using the `verl`[1] framework (Sheng et al., 2025), which provides a unified and well-tested training pipeline for reinforcement learning with large language models. We adopt the official implementations of GRPO and its variants available in `verl`, and build our method on top of the same infrastructure. Specifically, we add lambda optimization process before calculate advantage, and modify the advantage estimation and policy update components where required.

**Algorithmic variants.** We directly use the built-in implementations of baselines provided by the `verl` without any modification. These methods share the same training pipeline, rollout configuration, optimization schedule, and infrastructure settings. The only difference among these algorithms lies in the computation of advantages from trajectory-level rewards. GRPO-style methods employ their respective group-based or generalized advantage formulations, while our method replaces this component with a dual-based prompt-wise advantage derived from an exact KL-constrained optimization.

**Hyperparameters.** Unless otherwise specified, we follow the default hyperparameter settings of the `verl`. This includes optimization parameters, batching strategies, rollout settings, and scheduling-related configurations. All compared methods are trained using the same learning rate, batch sizes, number of rollouts per query, and policy update epochs. No method-specific hyperparameter tuning is performed.

**KL regularization.** All methods employ an identical auxiliary KL loss between the policy and a fixed pretrained reference model, implemented using a low-variance estimator with coefficient $\beta = 0.001$. This term is used solely for numerical stability and reference anchoring.

**Model scale.** The majority of experiments and ablations are conducted using the Qwen2.5-Math-1.5B model, which allows for detailed and controlled analysis. To demonstrate scalability, we additionally report results on Qwen2.5-Math-7B. Due to computational cost, 7B experiments are limited to the main comparison tables and are not included in ablation studies.

**Reproducibility.** Table I summarizes all relevant hyperparameters. Except for the algorithm-specific components, all settings are shared across methods. This design ensures that observed performance differences can be attributed solely to differences in prompt-wise trust-region control.

| Category | Hyperparameter | GSPO | Ours |
|---|---|---|---|
| *Optimization & Training* | Learning Rate (LR) | $1 \times 10^{-6}$ (default), $1 \times 10^{-5}$ | |
| | Optimizer | AdamW | |
| | Weight Decay | 0.0 | |
| | PPO Update Epochs | 1 (default), 2, 5 | |
| *Batching Strategy* | Global Batch Size | 256 | |
| | PPO Mini-batch Size | 64 | |
| | Micro-batch Size (per GPU) | 8 | |
| *RL Algorithm* | Rollouts per Query ($N$) | 8 | |
| | Advantage Estimator | Group-normalized | Dual (prompt-wise) |
| | Clipping | $\varepsilon = 0.2$ | **None** |
| | KL Constraint | Not enforced | $\delta \in \{0.1, 0.01, 0.001\}$ |
| | Auxiliary KL Loss (policy vs ref) | $\beta = 0.001$ | |
| | Entropy Coefficient | 0.0 | |
| *Generation & Tokenization* | Rollout Temperature | 1.0 | |
| | Max Response Tokens | 1024 (1.5B), 3072 (7B) | |
| *Infrastructure & Scheduling* | Nodes | 1 | |
| | GPUs per Node | H200×2 (1.5B), H200×4 (7B / RM) | |
| | vLLM GPU Memory Utilization | 0.8 | |

*Table I.* Comparison of key hyperparameters between GSPO and our prompt-wise trust-region optimization. All non-algorithmic settings are shared unless otherwise specified.

---

[1] https://github.com/volcengine/verl

# C. Additional Results and Analysis

## C.1. Our Method Behavior and Interpretation

**Prompt-wise Update Magnitude.** Figure I-left provides a conceptual view of how update strength is allocated across prompts under our method. Using the learned trust-region parameters, we compute the resulting advantages as a function of the number of correct rollouts. As shown in the figure, prompts associated with fewer correct rollouts receive larger update magnitudes, while those with many correct rollouts are updated more conservatively. This illustrates how update effort is distributed across prompts once the trust-region scaling is in effect. Moreover, tighter trust-region budgets (smaller $\delta$) yield more uniformly attenuated update magnitudes across all values of $c$. This reflects the consequence of stronger regularization induced by larger $\lambda$ values, while preserving the relative ordering across prompts.

**Effect of $\delta$ on $\lambda$ Scaling.** Figure I-right shows the actual evolution of the prompt-wise Lagrange multiplier $\lambda$ during training under different trust-region budgets $\delta$. Smaller values of $\delta$ consistently lead to larger $\lambda$, corresponding to stronger penalties on policy deviation. Notably, $\lambda$ evolves smoothly over the course of training and stabilizes without abrupt oscillations or collapse, indicating that trust-region strength is adaptively learned rather than manually scheduled.

Taken together, these two views connect training dynamics and resulting behavior: the right panel demonstrates how $\lambda$ is formed during training, while the left panel illustrates the update allocation behavior implied by the learned $\lambda$ values. This relationship highlights how the trust-region budget $\delta$ controls update behavior indirectly through learned prompt-wise regularization, without relying on heuristic clipping or fixed regularization coefficients.

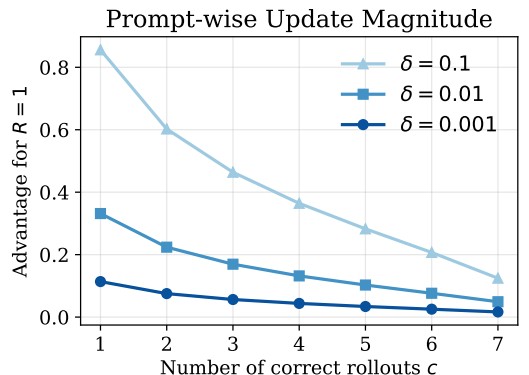
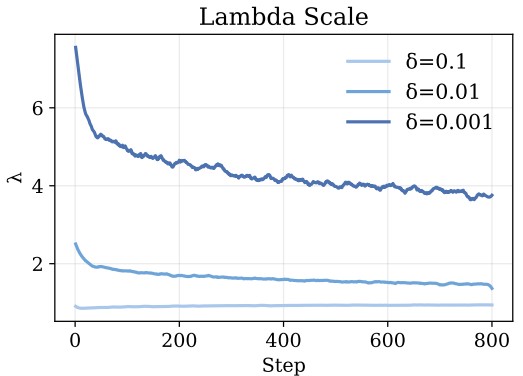

**Prompt-wise update magnitude.** Update strength decreases as the number of correct rollouts $c$ increases, with tighter trust regions (smaller $\delta$) yielding more conservative updates.

**Effect of $\delta$ on $\lambda$ scaling.** Smaller trust-region budgets $\delta$ lead to larger prompt-wise Lagrange multipliers $\lambda$, inducing stronger regularization against large policy updates.

*Figure I.* Behavioral analysis of our prompt-wise trust-region mechanism.

## C.2. Policy Entropy

Token-level policy entropy is used to quantify the diversity of the model's output distribution during training. For a given question $q$ and a sampled output $o^{(i)} = (o_1^{(i)}, \ldots, o_{|o^{(i)}|}^{(i)})$, the token-level policy entropy at time step $t$ is defined as the Shannon entropy of the next-token distribution:

$$H_t^{(i)} = -\sum_{v \in \mathcal{V}} \pi_\theta\left(v \mid q, o_{<t}^{(i)}\right) \log \pi_\theta\left(v \mid q, o_{<t}^{(i)}\right),$$

where $\mathcal{V}$ denotes the vocabulary. Entropy values are averaged over tokens and used to analyze training dynamics and policy diversity.

## C.3. Dataset-wise Pass@$k$ Results

We reports detailed dataset-wise Pass@$k$ results for all benchmarks Tab. II), evaluated at $k \in \{1, 2, 4, 8, 16, 32, 64, 128, 256\}$. While the main paper focuses on the overall average performance across datasets in Tab. 1 to highlight general trends, this appendix provides per-dataset visualizations and numerical results to support fine-grained analysis and reproducibility. Across all benchmarks, our method consistently outperforms the baselines, with particularly notable improvements in the high-$k$ regime. In addition, the performance gap between our method and prior GRPO-style baselines is more pronounced on certain datasets, such as AMC23 and AIME25.

*Table II.* Pass@$k$ on six mathematical reasoning benchmarks.

### MATH500

| Method | 1 | 2 | 4 | 8 | 16 | 32 | 64 | 128 | 256 |
|---|---|---|---|---|---|---|---|---|---|
| **Qwen2.5-Math-1.5B** | | | | | | | | | |
| GRPO | **69.97** | **75.23** | 79.04 | 82.02 | 84.43 | 86.30 | 87.68 | 88.81 | 90.00 |
| GPG | 68.97 | 74.66 | 78.96 | 82.29 | 84.78 | 86.58 | 87.96 | 89.19 | 90.40 |
| GMPO | 69.28 | 74.37 | 78.20 | 81.25 | 83.75 | 85.74 | 87.17 | 88.26 | 89.60 |
| GSPO | 69.10 | 74.25 | 78.12 | 81.16 | 83.57 | 85.49 | 86.97 | 88.09 | 89.00 |
| DAPO | 69.17 | 74.04 | 77.80 | 81.10 | 83.96 | 86.25 | 87.93 | 89.23 | 90.40 |
| CISPO | 69.45 | 74.57 | 78.50 | 81.56 | 84.02 | 86.09 | 87.81 | 89.09 | 90.00 |
| Ours $\delta=0.1$ | 69.70 | 75.19 | **79.28** | 82.42 | 84.96 | 87.02 | 88.63 | 90.00 | 91.40 |
| Ours $\delta=0.01$ | 67.84 | 74.30 | 78.93 | 82.46 | 85.30 | **87.85** | 89.41 | 90.92 | 92.00 |
| Ours $\delta=0.001$ | 65.84 | 73.30 | 78.63 | **82.50** | **85.47** | 87.74 | **89.51** | **91.03** | 92.40 |
| **Qwen2.5-Math-7B** | | | | | | | | | |
| GSPO | 72.20 | 78.18 | 82.21 | 85.02 | 87.13 | 88.80 | 90.12 | 91.20 | 92.00 |
| Ours $\delta=0.01$ | **72.95** | **79.19** | **83.33** | **86.24** | **88.34** | **89.85** | **91.07** | 92.05 | 92.80 |
| Ours $\delta=0.001$ | 67.11 | 75.33 | 80.94 | 84.67 | 87.38 | 89.48 | 91.01 | **92.21** | **93.20** |

### AMC23

| Method | 1 | 2 | 4 | 8 | 16 | 32 | 64 | 128 | 256 |
|---|---|---|---|---|---|---|---|---|---|
| **Qwen2.5-Math-1.5B** | | | | | | | | | |
| GRPO | 50.91 | 61.29 | 69.21 | 74.50 | 78.29 | 81.20 | 83.73 | 85.93 | 87.50 |
| GPG | 51.63 | 62.39 | 70.82 | 76.33 | 79.82 | 82.75 | 85.72 | 88.29 | 90.00 |
| GMPO | 52.31 | 62.07 | 70.21 | 76.13 | 79.85 | 82.38 | 84.90 | 86.79 | 87.50 |
| GSPO | 51.47 | 61.00 | 69.25 | 76.58 | 82.33 | 85.62 | 87.01 | 87.46 | 87.50 |
| DAPO | 51.98 | 61.63 | 69.38 | 74.59 | 78.27 | 81.67 | 85.15 | 89.22 | 95.00 |
| CISPO | 50.43 | 59.58 | 67.07 | 72.82 | 77.30 | 80.97 | 84.46 | 87.75 | 90.00 |
| Ours $\delta=0.1$ | **53.44** | **63.89** | **71.58** | 76.92 | 81.25 | 85.19 | 88.52 | 91.09 | 92.50 |
| Ours $\delta=0.01$ | 49.79 | 61.33 | 71.03 | **78.27** | **83.28** | **87.21** | 90.89 | 93.68 | 95.00 |
| Ours $\delta=0.001$ | 48.94 | 59.66 | 68.38 | 75.41 | 81.37 | 86.63 | **90.99** | **94.29** | **97.50** |
| **Qwen2.5-Math-7B** | | | | | | | | | |
| GSPO | **66.56** | **72.72** | 76.36 | 78.70 | 80.34 | 82.16 | 84.20 | 86.02 | 87.50 |
| Ours $\delta=0.01$ | 63.18 | 70.25 | 74.90 | 78.41 | 81.78 | 85.74 | 90.44 | 94.81 | **97.50** |
| Ours $\delta=0.001$ | 63.37 | 71.44 | **76.39** | **79.70** | **82.91** | **86.60** | **90.89** | **95.19** | **97.50** |

### AIME24

| Method | 1 | 2 | 4 | 8 | 16 | 32 | 64 | 128 | 256 |
|---|---|---|---|---|---|---|---|---|---|
| **Qwen2.5-Math-1.5B** | | | | | | | | | |
| GRPO | 13.62 | 18.50 | 22.86 | 26.85 | 30.59 | 34.73 | 40.21 | 46.99 | 53.33 |
| GPG | **16.38** | 20.63 | 23.92 | 27.30 | 30.70 | 34.46 | 39.42 | 46.68 | 56.67 |
| GMPO | 14.13 | 19.29 | 24.29 | **29.20** | **33.38** | 37.40 | 42.65 | **49.89** | **60.00** |
| GSPO | 16.28 | **20.97** | **25.00** | 28.50 | 31.52 | 34.58 | 38.31 | 43.32 | 50.00 |
| DAPO | 14.99 | 19.28 | 22.55 | 25.99 | 29.63 | 32.87 | 36.25 | 41.67 | 50.00 |
| CISPO | 12.92 | 17.19 | 21.11 | 25.41 | 30.02 | 34.82 | 40.45 | 47.52 | 53.33 |
| Ours $\delta=0.1$ | 16.13 | 20.28 | 23.26 | 26.00 | 29.14 | 32.54 | 36.08 | 40.01 | 43.33 |
| Ours $\delta=0.01$ | 13.62 | 18.01 | 22.10 | 26.75 | 31.71 | 36.43 | 41.30 | 47.09 | 53.33 |
| Ours $\delta=0.001$ | 12.28 | 17.74 | 22.74 | 27.51 | 32.72 | **37.98** | **42.70** | 48.34 | 56.67 |
| **Qwen2.5-Math-7B** | | | | | | | | | |
| GSPO | 31.54 | 36.80 | 41.39 | 45.73 | 49.85 | 53.62 | 56.66 | 59.58 | 63.33 |
| Ours $\delta=0.01$ | **32.88** | **39.85** | **45.34** | **49.81** | 53.56 | 56.90 | 60.30 | 64.49 | 70.00 |
| Ours $\delta=0.001$ | 28.93 | 36.24 | 42.25 | 48.15 | **53.91** | **58.93** | **63.61** | **69.07** | **76.67** |

### AIME25

| Method | 1 | 2 | 4 | 8 | 16 | 32 | 64 | 128 | 256 |
|---|---|---|---|---|---|---|---|---|---|
| **Qwen2.5-Math-1.5B** | | | | | | | | | |
| GRPO | 4.60 | 7.93 | 12.30 | 16.75 | 20.91 | 25.42 | 30.65 | 37.50 | 46.67 |
| GPG | 4.61 | 8.00 | 12.48 | 17.01 | 20.83 | 24.70 | 29.73 | 35.64 | 40.00 |
| GMPO | **6.67** | **10.69** | **14.94** | **18.45** | 21.58 | 24.91 | 28.52 | 32.50 | 36.67 |
| GSPO | 5.73 | 9.33 | 13.29 | 16.55 | 19.52 | 22.98 | 26.18 | 28.22 | 30.00 |
| DAPO | 4.04 | 6.92 | 10.61 | 14.25 | 17.26 | 19.89 | 22.91 | 27.30 | 33.33 |
| CISPO | 5.49 | 9.15 | 13.53 | 17.75 | 21.54 | 25.23 | 29.35 | 34.77 | 43.33 |
| Ours $\delta=0.1$ | 5.56 | 9.07 | 13.13 | 17.01 | 20.71 | 24.98 | 30.71 | 37.26 | 43.33 |
| Ours $\delta=0.01$ | 5.76 | 9.34 | 13.34 | 16.82 | 19.86 | 23.05 | 26.89 | 32.92 | 43.33 |
| Ours $\delta=0.001$ | 5.17 | 8.83 | 13.55 | 18.38 | **22.65** | **27.47** | **33.87** | **41.35** | **50.00** |
| **Qwen2.5-Math-7B** | | | | | | | | | |
| GSPO | 10.21 | 14.45 | 17.99 | 21.12 | 24.08 | 26.42 | 28.74 | 31.26 | 33.33 |
| Ours $\delta=0.01$ | **11.46** | **16.28** | **20.90** | **25.34** | **29.74** | **33.66** | 37.10 | 40.53 | 43.33 |
| Ours $\delta=0.001$ | 8.84 | 12.92 | 17.30 | 22.12 | 27.31 | 32.52 | **37.79** | **42.62** | **46.67** |

### MinervaMath

| Method | 1 | 2 | 4 | 8 | 16 | 32 | 64 | 128 | 256 |
|---|---|---|---|---|---|---|---|---|---|
| **Qwen2.5-Math-1.5B** | | | | | | | | | |
| GRPO | 18.73 | 22.84 | 26.52 | 29.91 | 33.16 | 36.33 | 39.40 | 42.23 | 44.85 |
| GPG | 18.76 | 22.94 | 26.70 | 30.13 | 33.38 | 36.59 | 39.72 | 42.37 | 44.49 |
| GMPO | 18.67 | 22.73 | 26.41 | 29.92 | 33.22 | 36.24 | 39.20 | 42.40 | 45.96 |
| GSPO | 18.75 | 22.75 | 26.41 | 29.95 | 33.43 | 36.81 | 39.92 | 42.71 | 45.96 |
| DAPO | 18.92 | 22.93 | 26.61 | 30.10 | 33.43 | 36.55 | 39.58 | 42.71 | 45.59 |
| CISPO | 19.01 | 22.98 | 26.48 | 29.72 | 33.05 | 36.63 | 40.16 | 43.39 | 46.32 |
| Ours $\delta=0.1$ | **18.95** | **23.32** | **27.21** | **30.69** | 33.83 | 36.91 | 40.14 | 43.35 | 46.32 |
| Ours $\delta=0.01$ | 17.92 | 22.52 | 26.69 | 30.48 | 33.98 | 37.35 | 40.51 | 43.26 | 45.96 |
| Ours $\delta=0.001$ | 16.79 | 21.45 | 25.96 | 30.28 | **34.33** | **38.10** | **41.71** | **44.99** | **47.43** |
| **Qwen2.5-Math-7B** | | | | | | | | | |
| GSPO | 22.63 | 26.85 | 30.80 | 34.72 | 38.79 | 42.65 | 45.86 | 48.39 | 50.37 |
| Ours $\delta=0.01$ | **22.74** | **27.50** | **31.86** | 36.04 | 40.22 | 44.35 | 48.35 | **52.03** | **55.15** |
| Ours $\delta=0.001$ | 22.21 | 27.12 | 31.71 | **36.14** | **40.60** | **44.96** | **48.81** | 51.99 | 54.78 |

### OlympiadBench

| Method | 1 | 2 | 4 | 8 | 16 | 32 | 64 | 128 | 256 |
|---|---|---|---|---|---|---|---|---|---|
| **Qwen2.5-Math-1.5B** | | | | | | | | | |
| GRPO | 32.11 | 37.86 | 42.98 | **47.73** | 51.33 | 54.83 | 58.02 | 61.03 | 63.80 |
| GPG | 31.80 | 37.63 | 42.88 | 47.54 | 51.68 | 55.37 | 58.53 | 61.23 | 63.95 |
| GMPO | 31.96 | 37.58 | 42.60 | 47.10 | 51.13 | 54.68 | 57.86 | 60.77 | 63.35 |
| GSPO | 31.56 | 37.08 | 42.06 | 46.46 | 50.39 | 53.96 | 57.13 | 59.96 | 62.61 |
| DAPO | 31.98 | 37.48 | 42.41 | 46.91 | 50.98 | 54.52 | 57.58 | 60.54 | 63.80 |
| CISPO | **32.28** | **38.01** | **43.12** | 47.59 | 51.46 | 54.76 | 57.58 | 60.23 | 62.91 |
| Ours $\delta=0.1$ | 32.02 | 37.80 | 42.93 | 47.45 | 51.47 | 55.15 | 58.64 | 61.96 | 64.99 |
| Ours $\delta=0.01$ | 31.17 | 37.20 | 42.62 | 47.48 | 51.98 | 56.13 | 59.68 | 62.60 | 65.13 |
| Ours $\delta=0.001$ | 29.36 | 36.08 | 42.20 | 47.67 | **52.57** | **56.99** | **60.95** | **64.58** | **68.10** |
| **Qwen2.5-Math-7B** | | | | | | | | | |
| GSPO | 37.17 | 43.02 | 48.42 | 53.21 | 57.25 | 60.48 | 63.12 | 65.45 | 67.66 |
| Ours $\delta=0.01$ | **37.27** | **43.78** | **49.67** | 54.79 | 58.99 | 62.59 | 66.01 | 69.15 | 71.51 |
| Ours $\delta=0.001$ | 36.48 | 43.24 | 49.45 | **54.93** | **59.41** | **63.01** | **66.15** | 69.05 | **71.66** |

## C.4. Unique Correct Count (UCC) @ small $k$

In this section, we report UCC@k results for small values of $k$ ($k = 8, 16$) across datasets. While UCC@k is designed to measure the presence of multiple distinct correct solution modes, its interpretation becomes reliable only when a sufficiently large number of samples is available. At small $k$, the number of sampled responses is often insufficient to reveal multiple modes even when they exist, causing UCC@k to largely reflect correctness rather than diversity. We therefore focus on larger $k$ ($k \geq 32$) in the main paper, where diversity becomes meaningfully observable. The results reported here are provided for completeness and transparency, and they are consistent with the trends observed at larger $k$.

*Table III.* UCC@$k$ ($k = 8, 16$) across six mathematical reasoning benchmarks (MATH500, AMC23, AIME24, AIME25, MinervaMath, and OlympiadBench).

| Dataset | 8 | | | | | 16 | | | | |
|---|---|---|---|---|---|---|---|---|---|---|
| | GRPO | GPG | GMPO | GSPO | Ours | GRPO | GPG | GMPO | GSPO | Ours |
| MATH500 | 1.77 | 3.09 | 1.95 | 1.92 | **3.50** | 2.65 | 5.73 | 3.16 | 3.12 | **6.78** |
| AMC23 | 1.96 | 3.31 | 2.32 | 2.38 | **3.51** | 3.21 | 6.21 | 3.96 | 4.11 | **6.79** |
| AIME24 | 0.66 | 0.97 | 0.87 | 0.77 | **0.97** | 1.11 | 1.89 | 1.56 | 1.37 | **1.90** |
| AIME25 | 0.30 | 0.43 | 0.40 | 0.38 | **0.35** | 0.49 | 0.81 | 0.74 | 0.69 | **0.69** |
| MinervaMath | 0.69 | 0.84 | 0.71 | 0.74 | **0.86** | 1.20 | 1.58 | 1.27 | 1.32 | **1.67** |
| OlympiadBench | 1.01 | 1.41 | 1.07 | 1.11 | **1.53** | 1.72 | 2.64 | 1.88 | 1.96 | **2.98** |

## C.5. Output Diversity Analysis

**Similarity under Different Metrics** We measure output similarity using four complementary metrics, each capturing a different notion of overlap between generated responses. These metrics can be used both to define near-duplicate criteria when clustering correct outputs for **UCC@k** computation, and to directly quantify output diversity at the surface-form level.

- **TF-IDF Cosine Similarity.** Each response is represented as a TF-IDF weighted bag-of-words vector, and similarity is computed using cosine similarity. By down-weighting common terms and emphasizing discriminative words, TF-IDF focuses on lexical content overlap rather than exact phrasing. This metric captures whether two responses rely on similar key concepts or reasoning tokens.

- **ROUGE-L.** ROUGE-L measures similarity based on the length of the longest common subsequence (LCS) between two responses. Unlike n-gram overlap, it is tolerant to insertions and reordering, making it sensitive to shared sentence structure or reasoning flow. Higher ROUGE-L indicates that two responses follow similar solution trajectories.

- **Jaccard Similarity.** Jaccard similarity computes the ratio between the intersection and union of token sets. This metric ignores word order and frequency, focusing solely on whether the same lexical items appear in both responses. It provides a coarse but intuitive measure of content overlap.

- **Normalized Edit Similarity.** Edit similarity is derived from the normalized Levenshtein distance, measuring the minimum number of character-level insertions, deletions, and substitutions required to transform one response into another. This metric is highly sensitive to surface-form variations, capturing fine-grained differences such as formatting, numerical changes, or minor textual edits.

Across all metrics, our method consistently exhibits lower similarity, indicating a broader coverage of distinct correct solution patterns.

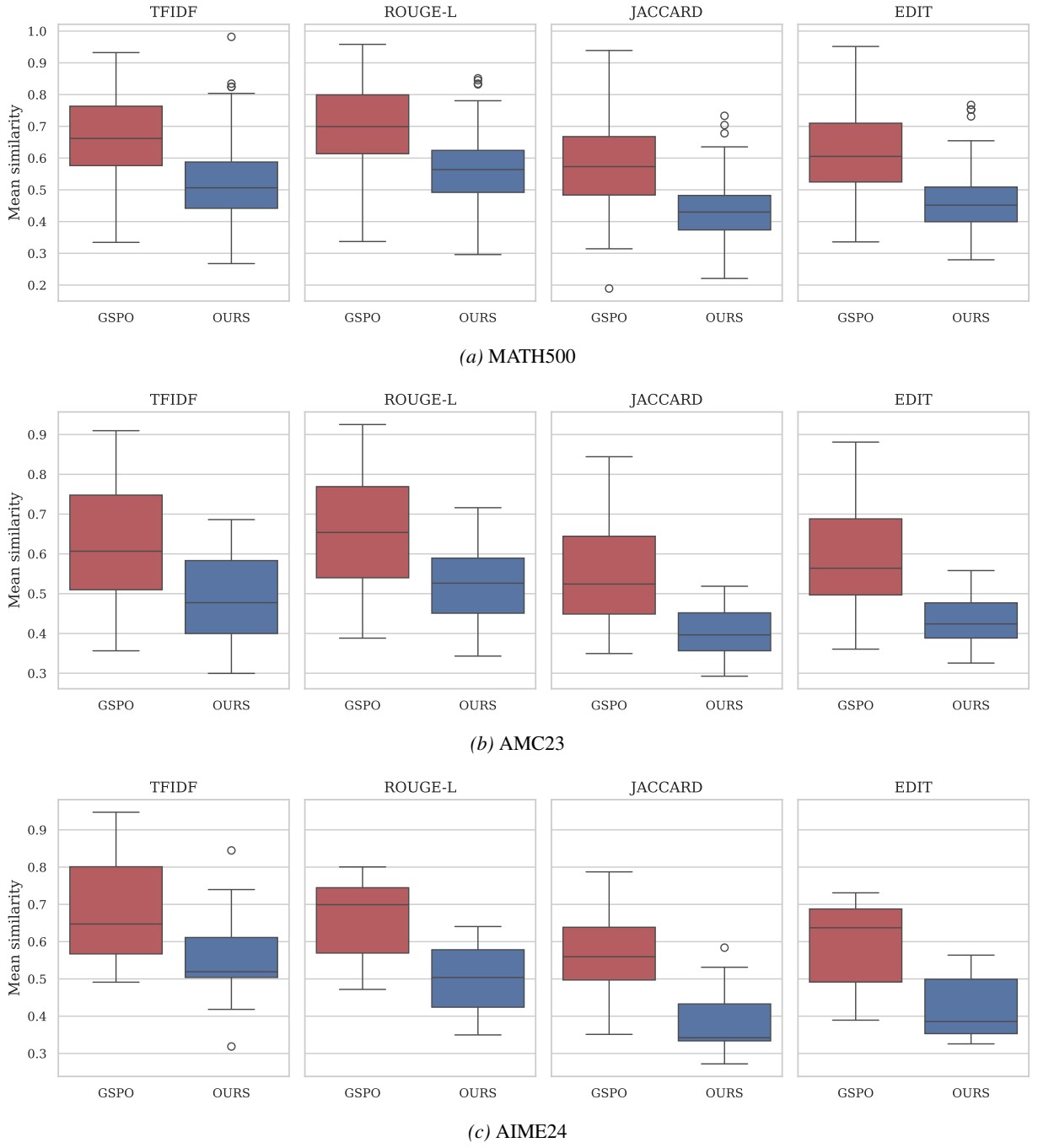

*(a)* MATH500

*(b)* AMC23

*(c)* AIME24

*Figure II.* Output similarity distributions under different metrics (TF-IDF, ROUGE-L, Jaccard, and Edit similarity) for three representative datasets. Lower similarity indicates higher output diversity.

### C.6. Qualitative Examples

We randomly sample responses from 256 rollouts with identical decoding parameters. Despite random sampling, GSPO exhibits strong trajectory collapse, repeatedly producing near-duplicate reasoning traces that are indistinguishable even at a superficial level. In contrast, our method yields genuinely diverse solution paths, reflecting multiple distinct constructions that satisfy the problem constraints. This diversity allows the model to maintain multiple viable reasoning paths, thereby increasing the likelihood of arriving at correct solutions. This qualitative observation is further supported by pairwise response similarity analysis (Figure III), which reveals uniformly high similarity for GSPO and substantially lower similarity for our method.

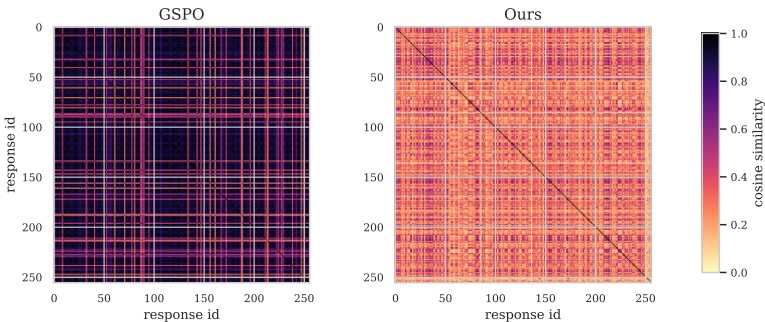

*Figure III.* Example of pairwise TF–IDF cosine similarity between 256 sampled responses.

---

**Question and Ground Truth**

**Question.** For how many ordered pairs $(a, b)$ of integers does the polynomial

$$x^3 + ax^2 + bx + 6$$

have *three distinct integer roots*? Please reason step by step and report the final answer in a boxed form.
**Ground Truth.**

$$\boxed{5}$$

---

**Responses from GSPO (3 out of 256)**

**Sample 1.** We need to find the number of ordered pairs $(a, b)$ of integers such that the polynomial $x^3 + ax^2 + bx + 6$ has 3 distinct integer roots. If the polynomial $x^3 + ax^2 + bx + 6$ has roots $r$, $s$, and $t$, then by Vieta's formulas, we have: 1. $r + s + t = -a$ 2. $rs + rt + st = b$ 3. $rst = -6$ We need to find all sets of distinct integers $(r, s, t)$ whose product is $-6$. The possible sets of distinct integer roots are:

$$(1, 2, -3), (1, -2, 3), (-1, 2, 3), (-1, -2, -3), (1, -1, 6), (1, 1, -6), (-1, -1, 6), (-1, 1, -6)$$

But we can discard the sets with repeated roots or non-distinct roots. So the valid sets are:

$$(1, 2, -3), (1, -2, 3), (-1, 2, 3), (-1, -2, -3)$$

For each set of roots, we can compute $a$ and $b$: 1. For $(1, 2, -3)$:

$$a = -(1 + 2 - 3) = 0, \quad b = 1 \cdot 2 + 1 \cdot (-3) + 2 \cdot (-3) = 2 - 3 - 6 = -7$$

So, $(a, b) = (0, -7)$. 2. For $(1, -2, 3)$:

$$a = -(1 - 2 + 3) = -2, \quad b = 1 \cdot (-2) + 1 \cdot 3 + (-2) \cdot 3 = -2 + 3 - 6 = -5$$

So, $(a, b) = (-2, -5)$. 3. For $(-1, 2, 3)$:

$$a = -(-1 + 2 + 3) = -4, \quad b = (-1) \cdot 2 + (-1) \cdot 3 + 2 \cdot 3 = -2 - 3 + 6 = 1$$

So, $(a, b) = (-4, 1)$. 4. For $(-1, -2, -3)$:

$$a = -(-1 - 2 - 3) = 6, \quad b = (-1) \cdot (-2) + (-1) \cdot (-3) + (-2) \cdot (-3) = 2 + 3 + 6 = 11$$

So, $(a, b) = (6, 11)$. Thus, the distinct ordered pairs $(a, b)$ are $(0, -7), (-2, -5), (-4, 1)$, and $(6, 11)$. Therefore, there are 4 such pairs. The final answer is $\boxed{4}$.

**Sample 2.** We need to find the number of ordered pairs $(a, b)$ of integers such that the polynomial $x^3 + ax^2 + bx + 6$ has 3 distinct integer roots. If the polynomial $x^3 + ax^2 + bx + 6$ has roots $r, s$, and $t$, then by Vieta's formulas, we have: 1. $r + s + t = -a$ 2. $rs + rt + st = b$ 3. $rst = -6$ We need to find all sets of distinct integers $(r, s, t)$ whose product is $-6$. The possible sets of distinct integer roots $(r, s, t)$ are:

$$(1, 2, -3), (1, -2, 3), (-1, 2, 3), (-1, -2, -3), (1, -1, 6), (1, 1, -6), (-1, -1, 6), (-1, 1, -6)$$

However, we need to exclude the sets with repeated roots. So the valid sets are:

$$(1, 2, -3), (1, -2, 3), (-1, 2, 3), (-1, -2, -3)$$

For each set of roots, we can calculate $a$ and $b$: 1. For $(1, 2, -3)$:

$$a = -(1 + 2 - 3) = 0, \quad b = 1 \cdot 2 + 1 \cdot (-3) + 2 \cdot (-3) = 2 - 3 - 6 = -7$$

So, $(a, b) = (0, -7)$. 2. For $(1, -2, 3)$:

$$a = -(1 - 2 + 3) = -2, \quad b = 1 \cdot (-2) + 1 \cdot 3 + (-2) \cdot 3 = -2 + 3 - 6 = -5$$

So, $(a, b) = (-2, -5)$. 3. For $(-1, 2, 3)$:

$$a = -(-1 + 2 + 3) = -4, \quad b = (-1) \cdot 2 + (-1) \cdot 3 + 2 \cdot 3 = -2 - 3 + 6 = 1$$

So, $(a, b) = (-4, 1)$. 4. For $(-1, -2, -3)$:

$$a = -(-1 - 2 - 3) = 6, \quad b = (-1) \cdot (-2) + (-1) \cdot (-3) + (-2) \cdot (-3) = 2 + 3 + 6 = 11$$

So, $(a, b) = (6, 11)$. Thus, the ordered pairs $(a, b)$ are $(0, -7), (-2, -5), (-4, 1)$, and $(6, 11)$. There are 4 such pairs. The final answer is:

$$\boxed{4}$$

**Sample 3.** We need to find the number of ordered pairs $(a, b)$ of integers such that the polynomial $x^3 + ax^2 + bx + 6$ has 3 distinct integer roots. If the polynomial $x^3 + ax^2 + bx + 6$ has roots $r, s, t$, then by Vieta's formulas, we have: 1. $r + s + t = -a$ 2. $rs + rt + st = b$ 3. $rst = -6$ We need to find all sets of distinct integers $(r, s, t)$ whose product is $-6$. The possible sets of $(r, s, t)$ are:

$$(1, 2, -3), (1, -2, 3), (-1, 2, 3), (-1, -2, -3), (1, -1, 6), (1, 1, -6), (-1, -1, 6), (-1, 1, -6)$$

But we only consider the sets with distinct integers, so we have:

$$(1, 2, -3), (1, -2, 3), (-1, 2, 3), (-1, -2, -3)$$

For each set, we can calculate $a$ and $b$: 1. For $(1, 2, -3)$:

$$a = -(1 + 2 - 3) = 0, \quad b = 1 \cdot 2 + 1 \cdot (-3) + 2 \cdot (-3) = 2 - 3 - 6 = -7$$

So, $(a, b) = (0, -7)$. 2. For $(1, -2, 3)$:

$$a = -(1 - 2 + 3) = -2, \quad b = 1 \cdot (-2) + 1 \cdot 3 + (-2) \cdot 3 = -2 + 3 - 6 = -5$$

So, $(a, b) = (-2, -5)$. 3. For $(-1, 2, 3)$:

$$a = -(-1 + 2 + 3) = -4, \quad b = (-1) \cdot 2 + (-1) \cdot 3 + 2 \cdot 3 = -2 - 3 + 6 = 1$$

So, $(a, b) = (-4, 1)$. 4. For $(-1, -2, -3)$:

$$a = -(-1 - 2 - 3) = 6, \quad b = (-1) \cdot (-2) + (-1) \cdot (-3) + (-2) \cdot (-3) = 2 + 3 + 6 = 11$$

So, $(a, b) = (6, 11)$. Thus, the distinct ordered pairs $(a, b)$ are $(0, -7)$, $(-2, -5)$, $(-4, 1)$, and $(6, 11)$. Therefore, there are 4 such pairs. The final answer is $\boxed{4}$.

---

**Responses from Ours (3 out of 256)**

**Sample 1.** We'll start by considering the polynomial $P(x) = x^3 + ax^2 + bx + 6$. If $r_1, r_2, r_3$ are the three distinct integer roots of the polynomial, then by Vieta's formulas, we have:
1. $r_1 + r_2 + r_3 = -a$ 2. $r_1 r_2 + r_2 r_3 + r_3 r_1 = b$ 3. $r_1 r_2 r_3 = -6$
We need to find all sets of distinct integers $(r_1, r_2, r_3)$ whose product is $-6$. The possible sets of distinct integer roots $(r_1, r_2, r_3)$ are permutations of $(1, -1, 6)$, $(1, -1, -6)$, $(1, 2, -3)$, $(1, -2, 3)$, $(-1, 2, 3)$, and $(-1, -2, -3)$. For each set, we can calculate $a$ and $b$ using Vieta's formulas.
Let's enumerate all the possible sets and calculate the corresponding $a$ and $b$ values.
1. For $(1, -1, 6)$:

$$a = -(1 - 1 + 6) = -6, \quad b = 1(-1) + (-1)(6) + 6(1) = -1 - 6 + 6 = -1$$

So, $(a, b) = (-6, -1)$.
2. For $(1, -1, -6)$:

$$a = -(1 - 1 - 6) = 6, \quad b = 1(-1) + (-1)(-6) + (-6)(1) = -1 + 6 - 6 = -1$$

So, $(a, b) = (6, -1)$.
3. For $(1, 2, -3)$:

$$a = -(1 + 2 - 3) = 0, \quad b = 1(2) + 2(-3) + (-3)(1) = 2 - 6 - 3 = -7$$

So, $(a, b) = (0, -7)$.
4. For $(1, -2, 3)$:

$$a = -(1 - 2 + 3) = -2, \quad b = 1(-2) + (-2)(3) + 3(1) = -2 - 6 + 3 = -5$$

So, $(a, b) = (-2, -5)$.
5. For $(-1, 2, 3)$:

$$a = -(-1 + 2 + 3) = -4, \quad b = (-1)(2) + 2(3) + 3(-1) = -2 + 6 - 3 = 1$$

So, $(a, b) = (-4, 1)$.
6. For $(-1, -2, -3)$:

$$a = -(-1 - 2 - 3) = 6, \quad b = (-1)(-2) + (-2)(-3) + (-3)(-1) = 2 + 6 + 3 = 11$$

So, $(a, b) = (6, 11)$.
We can see that each set of roots $(r_1, r_2, r_3)$ gives a unique pair $(a, b)$. Therefore, there are 6 distinct ordered pairs $(a, b)$. The answer is:

$$\boxed{6}$$

**Sample 2.** The polynomial $x^3 + ax^2 + bx + 6$ has 3 distinct integer roots, say $r_1, r_2, r_3$. By Vieta's formulas, we know: 1. $r_1 + r_2 + r_3 = -a$ 2. $r_1 r_2 + r_2 r_3 + r_3 r_1 = b$ 3. $r_1 r_2 r_3 = -6$

We need to find all possible sets of distinct integer roots $(r_1, r_2, r_3)$ whose product is $-6$. The integer factorizations of $-6$ into three distinct integers are:

$$(-1, 1, -6), (-1, 2, -3), (-2, 1, 3), (1, -2, 3)$$

For each set of roots, we can compute $a$ and $b$: 1. For $(-1, 1, -6)$:

$$a = -(-1 + 1 - 6) = 6, \quad b = (-1 \cdot 1) + (1 \cdot -6) + (-6 \cdot -1) = -1 - 6 + 6 = -1$$

So, $(a, b) = (6, -1)$.
2. For $(-1, 2, -3)$:

$$a = -(-1 + 2 - 3) = 2, \quad b = (-1 \cdot 2) + (2 \cdot -3) + (-3 \cdot -1) = -2 - 6 + 3 = -5$$

So, $(a, b) = (2, -5)$.
3. For $(-2, 1, 3)$:

$$a = -(-2 + 1 + 3) = -2, \quad b = (-2 \cdot 1) + (1 \cdot 3) + (3 \cdot -2) = -2 + 3 - 6 = -5$$

So, $(a, b) = (-2, -5)$.
4. For $(1, -2, 3)$:

$$a = -(1 - 2 + 3) = -2, \quad b = (1 \cdot -2) + (-2 \cdot 3) + (3 \cdot 1) = -2 - 6 + 3 = -5$$

So, $(a, b) = (-2, -5)$.
We see that the pairs $(6, -1)$, $(2, -5)$, and $(-2, -5)$ are distinct. So, the number of ordered pairs $(a, b)$ is 3.
Thus, the final answer is:

$$\boxed{3}$$

**Sample 3.** To solve this problem, we need to find the number of ordered pairs $(a, b)$ of integers such that the polynomial $x^3 + ax^2 + bx + 6$ has 3 distinct integer roots.
**Steps:**
1. **Polynomial Roots:** If $r_1, r_2, r_3$ are the distinct integer roots of the polynomial $x^3 + ax^2 + bx + 6$, then by Vieta's formulas, we have:

$$r_1 + r_2 + r_3 = -a$$
$$r_1 r_2 + r_2 r_3 + r_3 r_1 = b$$
$$r_1 r_2 r_3 = -6$$

2. **Finding Integer Root Combinations:** We need to find all distinct integer triplets $(r_1, r_2, r_3)$ whose product is $-6$. We then need to ensure that the sum $r_1 + r_2 + r_3$ is an integer and compute $a$ and $b$ accordingly.
3. **Counting Valid Pairs $(a, b)$:** For each valid triplet, compute $a$ and $b$ and ensure they are integers.
Let's write Python code to find all valid triplets and compute the corresponding $(a, b)$ pairs.

```
from itertools import permutations

# Function to find all distinct integer triplets whose product is -6
def find_triplets():
    triplets = []
    for r1 in range(-6, 7):
        for r2 in range(-6, 7):
```

```
            for r3 in range(-6, 7):
                if r1 * r2 * r3 == -6 and r1 != r2 \
                                    and r2 != r3 and r1 != r3:
                    triplets.append((r1, r2, r3))
    return triplets

# Function to compute (a, b) for each triplet
def compute_a_b(triplet):
    r1, r2, r3 = triplet
    a = -(r1 + r2 + r3)
    b = r1 * r2 + r2 * r3 + r3 * r1
    return a, b

# Find all triplets and compute (a, b) pairs
triplets = find_triplets()
ab_pairs = set(compute_a_b(triplet) for triplet in triplets)

# Count the number of unique (a, b) pairs
num_pairs = len(ab_pairs)
print(num_pairs)
```
```output
5
```

The number of ordered pairs $(a, b)$ of integers such that the polynomial $x^3 + ax^2 + bx + 6$ has 3 distinct integer roots is $\boxed{5}$.

## C.7. Behavioral Case Studies

**Flip Rate Analysis**   To analyze performance across different difficulty regimes, we conduct a flip-rate analysis based on difficulty buckets defined by the base (pretrained) policy. Specifically, for each prompt, we collect 256 rollouts from the base policy and group prompts according to the number of correct responses using bins $[0, 8, 32, 128, 256]$, corresponding to *very hard*, *hard*, *medium*, and *easy* buckets. This construction ensures that difficulty is determined independently of the fine-tuned models. Flip Rate is defined as the fraction of prompts whose number of correct responses becomes non-zero after fine-tuning, i.e., prompts that transition from having no correct solutions under the base policy to at least one correct solution under the trained policy. As shown in Figure IV, our method consistently achieves higher flip rates in harder buckets, with the largest gains observed in the *very hard* and *hard* regimes. While the exact bucket boundaries may vary across datasets, the same qualitative trend holds: improvements at larger $k$ are driven by successfully solving previously unsolved and rare-success prompts, rather than repeatedly sampling problems that are already easy for the base model.

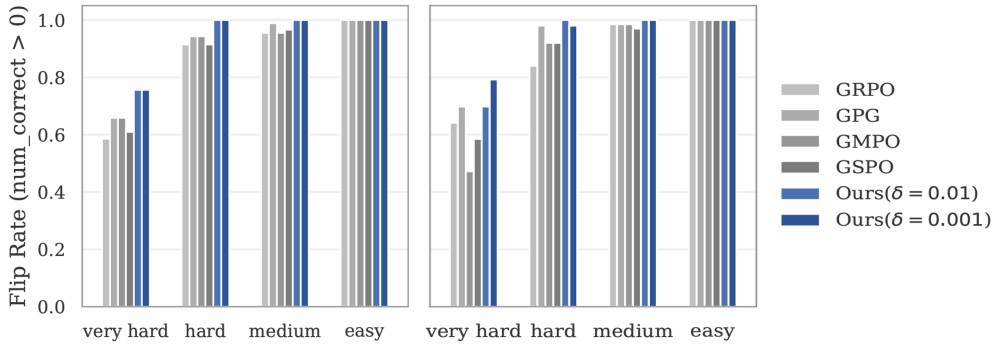

*Figure IV.* Flip rate across difficulty buckets on MATH500 (left) and OlympiadBench (right), measuring the fraction of prompts that transition from zero to non-zero correctness after fine-tuning.

**Temperature Variation**   A natural question is whether increasing the sampling temperature in GSPO can achieve comparable gains to our method, as higher temperature is known to increase sampling diversity and often improves Pass@$k$. To examine this, we evaluate GSPO under an elevated temperature setting ($\text{temp} = 1.5$) and compare it with our method using the Unique Correct Ratio (UCR), defined as the fraction of distinct correct solutions among 256 sampled responses. While higher temperature does lead to moderate improvements in UCR for GSPO, the gains remain consistently smaller than those achieved by our method across datasets. Moreover, temperature-based diversification exhibits diminishing returns and eventually degrades solution quality as temperature increases, reflecting a trade-off between randomness and correctness. In contrast, our method maintains a policy that explicitly preserves multiple correct solution modes, rather than relying on increased sampling noise. This results in consistently higher UCR, indicating that our diversity gains stem from structured mode coverage rather than stochastic perturbations at inference time.

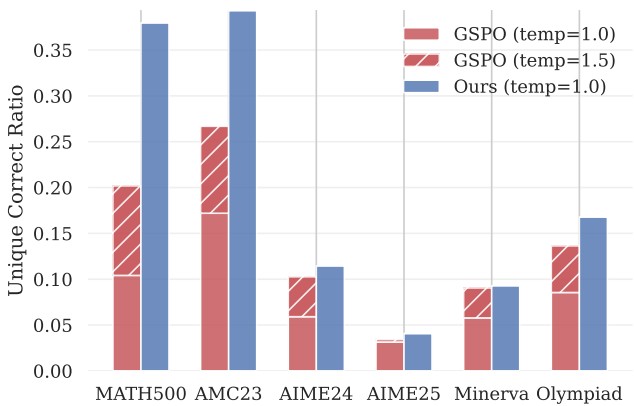

*Figure V.* Unique correct ratio across datasets under different sampling temperatures. GSPO benefits from increased temperature, while our method maintains higher ratios.

