# OpenReview forum: "QUATRO: Query-Adaptive Trust Region Policy Optimization for LLM Fine-tuning"
_ICML.cc/2026/Conference — ICML 2026 regular_

### Official Review · Reviewer_9SuM · 2026-03-12

**Soundness:** 3
**Presentation:** 3
**Significance:** 3
**Originality:** 3
**Overall Recommendation:** 4
**Confidence:** 4

**Summary:**

This paper proposes QUATRO that directly enforces trust-region constraints through a principled optimization. The authors identify significant limitations in existing reinforcement learning RL methods, such as GSPO, which rely on heuristic approaches to regulate policy updates. The experimental results demonstrate that the proposed method achieves performance improvements compared to the baselines.

**Compliance With Llm Reviewing Policy:**

Affirmed.

**Final Justification:**

My concerns have been fully addressed, and I no longer have any questions. I will maintain my positive score.

**Key Questions For Authors:**

See Weakness.

**Limitations:**

The authors should add a clearer limitations section.

**Strengths And Weaknesses:**

Strengths：
1. The article is well-organized and solid in content.
2. The motivation of the article is clear, and the author clearly explains why constraints need to be added in QUATRO.
3. On some datasets, the performance of the proposed method significantly surpasses that of the baselines.

Weaknesses：
1. The author provides only the hyperparameters and does not offer clear source code for the review and reproducibility of the experiments. If the author can provide reproducible source code, I will consider raising my score.
2. The introduction of constraints brings additional overhead. In terms of time and space complexity, how much is the theoretical extra cost compared to the baselines?
3. The specific experimental conditions and setup of Figure 1 are not clearly defined.
4. Do the experimental phenomena and effectiveness still exist and remain valid in models other than Qwen, such as the Llama 3 series?

---

> ### Author Rebuttal · Authors · 2026-03-31
>
> We sincerely thank the reviewer for thoughtful and constructive feedback. In response, we clarify the experimental setup, discuss computational overhead more concretely, and expand the discussion of reproducibility, generalization, and limitations.
>
> ---
>
> __(W1) Lack of reproducible code__
>
> __A.__ We thank the reviewer for this important point. Although we are not able to provide a URL due to the double-blind policy at this point, we are preparing to release the full training and evaluation code upon acceptance, along with detailed configurations (hyperparameters, training and evaluation scripts) to ensure reproducibility.
>
> ---
>
> __(W2) Additional computational overhead__
>
> __A.__ The additional cost is minimal, as it only involves solving __a per-query scalar__ dual variable $\lambda$. This optimization uses already-sampled trajectories and computed reward statistics, without requiring additional rollouts or auxiliary model training. In practice, as in standard LLM RL, the dominant cost remains in rollout generation and log-probability evaluation. Our dual optimization is relatively lightweight.
>
> For reference, we provide our timing logs on the 1.5B model, verifying that the dominant costs are rollout generation and log-probability evaluation. The former takes approximately 50-90 seconds per step (around 50-70% of the total time), and the latter accounts for an additional 10-30 seconds (around 10-25% of the total time). In contrast, the remaining components, including our advantage computation and dual optimization, take only a few seconds and are thus relatively negligible. Overall, the total time is nearly identical to the baseline, with no notable difference, indicating that the dual optimization introduces minimal overhead.
>
> ---
>
> __(W3) Unclear experimental setup__
>
> __A.__ Figure 1 essentially uses the same setup as Figure 4. Specifically, (a) varies the learning rate from the default setting (1e-6) to a higher value (1e-5), and (b) increases the number of inner-loop updates (1, 2, 5). All other settings are kept identical to the standard GSPO configuration. We will clarify these details in the revised version.
>
> ---
>
> __(W4) Generalization across model families__
>
> __A.__ We agree that evaluation across multiple model families would further strengthen the results. Due to the short rebuttal period and limited computing resources, we were not able to conduct this experiment, but will add it in camera-ready.
>
> ---
>
> __(W5) Lack of limitations discussion__
>
> __A.__ We thank the reviewer for this helpful suggestion and will add a limitations section in the revised version. In particular, our method relies on the KL constraint parameter $\delta$, which determines how far the updated policy is allowed to move from the reference policy. This gives a principled way to control the trade-off between exploitation and exploration, but the question of how to select $\delta$ automatically, or to better understand which values work well in practice, deserves further investigation. We will clarify this limitation in the revision.

---

> > ### Author Rebuttal · Reviewer_9SuM · 2026-04-03
> >
> > I greatly appreciate the authors' rebuttal. My concerns have been fully addressed, and I no longer have any questions. I will maintain my positive score. If reviewer gu3U acknowledges that the authors' responses have adequately addressed the concerns regarding gaps between theory and practice, I will consider raising my score further to support the authors.

---

> > > ### Author Response · Authors · 2026-04-06
> > >
> > > We are greatly thankful for taking time to read and acknowledge our rebuttal, and for your mention for further support.
> > >
> > > We would kindly ask to score primarily based on your judgement in this case. The concern raised by Reviewer gu3U appears to stem from a misunderstanding of our notation rather than a fundamental theory-practice gap. We have made our best efforts to clarify this point through our previous responses.
> > >
> > > However, as of now, we have not received a follow-up response from the reviewer for several days, despite our attempts to resolve the confusion. Since the reviewer discussion phase is now underway, we wanted to respectfully bring this to your attention.
> > >
> > > Thank you again for your time and consideration.

---

### Official Review · Reviewer_XV8V · 2026-03-13

**Soundness:** 3
**Presentation:** 3
**Significance:** 3
**Originality:** 3
**Overall Recommendation:** 4
**Confidence:** 2

**Summary:**

This paper studies reinforcement learning for LLM fine-tuning through the lens of trust-region optimization. The authors argue that widely used GRPO-style methods rely on heuristic importance-ratio clipping and group-wise normalization, which can become brittle under policy staleness, aggressive learning rates, and repeated offline inner-loop updates. They further argue that using a uniform trust-region treatment across all queries is suboptimal because different prompts have different difficulty levels and uncertainty profiles.

To address this, the paper proposes QUATRO, a query-adaptive trust-region policy optimization method. The key idea is to formulate a prompt-wise KL-constrained optimization problem, derive the corresponding dual variables, and obtain an exact trust-region objective that adaptively controls update magnitudes on a per-query basis rather than through heuristic clipping. The resulting objective resembles GSPO-style optimization but introduces an intrinsic stabilizing term and per-query dual control.Empirically, the paper evaluates QUATRO on mathematical reasoning tasks using Qwen2.5-Math-1.5B and 7B models. Across six benchmarks, the method improves average Pass@k over GRPO-style baselines, with larger gains at higher k, and it also reports substantially improved UCC@k, a new metric intended to measure the diversity of distinct correct solution paths. The paper further presents analyses suggesting improved robustness to learning-rate increases and policy staleness, as well as more stable entropy throughout training.

**Compliance With Llm Reviewing Policy:**

Affirmed.

**Final Justification:**

I generally agree with the concerns raised by **Reviewer gu3U**, particularly regarding the theoretical gaps between the trajectory-level derivation and the proposed token-level algorithm, as well as the issue around the adjoint variable optimization in Eq. (19). However, I also find the paper **solid in other aspects**, including its motivation and empirical evaluation, so I remain **4: Weak accept**.

**Key Questions For Authors:**

1. How broadly does QUATRO generalize beyond mathematical reasoning?
2. What is the practical overhead of the query-adaptive dual optimization compared with GRPO/GSPO-style baselines?
3. How much of the gain comes from better exploration/diversity versus simply more conservative updates?

**Limitations:**

The paper includes an impact statement, but the limitations discussion is still a bit light. In particular, I would encourage the authors to more explicitly discuss:
+ the current validation being limited to mathematical reasoning benchmarks and Qwen2.5-Math models;
+ the additional complexity introduced by per-query dual control compared with simpler heuristics; and
+ the possibility that some benefits may depend on specific reward distributions or response diversity patterns in reasoning tasks. On societal impact, the current statement is brief and likely acceptable, but a slightly more concrete discussion of how improved RL fine-tuning could affect deployment of capable reasoning models would make the paper stronger.

**Strengths And Weaknesses:**

>**Strengths**

The paper is technically solid overall. The authors start from a clean prompt-wise KL-constrained formulation and derive a dual-based objective in a principled way, rather than introducing another heuristic clipping rule. The core method is mathematically motivated, and the high-level interpretation is appealing: easier or more stable queries can tolerate sharper updates, while noisier queries induce larger dual penalties and therefore more conservative updates. I also appreciate that the paper connects this interpretation to concrete quantities such as the learned per-query dual variable and prompt-wise update magnitude.

On the empirical side, the paper evaluates against several relevant baselines—GRPO, GPG, GMPO, and GSPO—on six mathematical reasoning benchmarks and two model sizes. The reported Pass@k gains are consistent, and the gains become more pronounced as k increases, which matches the paper’s claim that the method preserves a richer solution distribution. The additional robustness experiments on entropy stability, learning-rate sensitivity, and stale/offline inner-loop updates strengthen the case that the method is not merely getting a small benchmark-specific gain.

>**Weaknesses**

My main reservation is that, although the method is well motivated, the evidence is still concentrated in one domain: mathematical reasoning with Qwen2.5-Math models. This is a strong and relevant testbed, but the paper occasionally sounds broader than what is directly demonstrated. I would be more convinced if the authors either moderated the scope of their claims or added results on another domain such as code reasoning or a more general RLVR setup. Relatedly, some of the comparisons are against recent GRPO-style variants, but the practical landscape is moving quickly, and it would help to better position QUATRO against the strongest current trust-region or adaptive-regularization alternatives.

---

> ### Author Rebuttal · Authors · 2026-03-31
>
> We sincerely thank the reviewer for thoughtful and constructive feedback. In response, we provide additional clarification, new empirical comparisons, and a broader discussion of generalization, limitations, and societal impact to better position the contribution of our work.
>
> ---
>
> __(W1, Q1) Generalization beyond mathematical reasoning__
>
> __A.__ Following the reviewer’s suggestion, we conduct two additional experiments on the code generation domain. First, we train the Qwen2.5-Coder-1.5B on APPS and evaluate it on HumanEval(+) datasets. According to the results below, our method consistently outperforms the GSPO baseline.
>
> HumanEval
>
> |Method|Pass@1|Pass@4|Pass@8|Pass@16|Pass@32|Pass@64|
> |-|-|-|-|-|-|-|
> |GSPO|2.42|8.22|13.93|21.83|31.18|40.85|
> |Ours|**2.94**|**10.07**|**17.23**|**27.19**|**38.67**|**50.00**|
>
> HumanEval+
>
> |Method|Pass@1|Pass@4|Pass@8|Pass@16|Pass@32|Pass@64|
> |-|-|-|-|-|-|-|
> |GSPO|1.98|6.82|11.72|18.72|27.37|37.20|
> |Ours|**2.35**|**8.06**|**13.85**|**22.08**|**31.85**|**41.46**|
>
> Additionally, we conduct zero-shot evaluation on another code generation benchmark, MBPP(+). As seen in the table below, our method consistently maintains stronger performance from a reasoning perspective, with more pronounced gains at Pass@16, than the GSPO baseline.
>
> |Method|MBPP Pass@1|MBPP Pass@16|MBPP+ Pass@1|MBPP+ Pass@16|
> |-|-|-|-|-|
> |GSPO|5.41|44.71|4.81|40.62|
> |Ours|**6.19**|**49.47**|**5.39**|**43.39**|
>
> ---
>
> __(W2) Positioning against recent methods__
>
> __A.__ We conduct additional experiments with recent methods such as DAPO and CISPO. These methods still rely on clipping or other heuristic mechanisms, whereas our approach enforces a constraint in a more principled manner. Probably because of this difference, we observe that these baselines tend to become less stable under more aggressive settings, e.g., higher learning rates or increased inner-loop updates, while our method remains robust. (Please refer to our answer to W3 for the reviewer gu3U for detailed results.)
>
> ---
>
> __(Q2, W3) Computational overhead__
>
> __A.__ The additional cost is minimal, as it only involves solving __a per-query scalar__ dual variable $\lambda$. This optimization uses already-sampled trajectories and computed reward statistics, without requiring additional rollouts or auxiliary model training. In practice, as in standard LLM RL, the dominant cost remains in rollout generation and log-probability evaluation. The dual optimization is relatively lightweight.
>
> For reference, we provide our timing logs on the 1.5B model, verifying that the dominant costs are rollout generation and log-probability evaluation. The former takes approximately 50-90 seconds per step (around 50-70% of the total time), and the latter accounts for an additional 10-30 seconds (around 10-25% of the total time). In contrast, the remaining components, including our advantage computation and dual optimization, take only a few seconds and are thus relatively negligible. Overall, the total time is nearly identical to the baseline, with no notable difference, indicating that the dual optimization introduces minimal overhead.
>
> ---
>
> __(Q3) Gain from exploration vs conservativeness__
>
> __A.__ We respectfully argue that QUATRO plays a more critical role than simply making updates more conservative. Particularly, it modulates sharpness in updates by query-wise adaptive control. This improves the trade-off between diversity and stability, preventing collapse while preserving useful exploration. As shown in our experiments, QUATRO maintains higher entropy, achieves stronger gains in UCC@k and Pass@k at larger $k$, and remains stable even under more aggressive settings (e.g., higher learning rates or increased staleness). These results indicate that the improvements come not only from conservative updates, but also from better diversity preservation.
>
> ---
>
> __(L) Task dependence and societal impact__
>
> __A.__ We thank the reviewer for this thoughtful suggestion. We will strengthen the discussion by noting that some of the observed benefits may depend on task-specific properties, such as the reward distribution and the degree of response diversity in reasoning problems. In particular, we hypothesize that our method may be especially beneficial under continuous rewards, since it adapts the trust-region in a reward-aware manner rather than relying only on normalization by the reward standard deviation as in GRPO. A more systematic analysis across different reward distributions would be an important future direction. We will also make the societal impact discussion more concrete by clarifying that improved RL fine-tuning could support the deployment of more reliable reasoning models, while also increasing the importance of careful evaluation and responsible use in high-impact settings.

---

> > ### Author Rebuttal · Reviewer_XV8V · 2026-04-03
> >
> > The authors have addressed my concerns well in the rebuttal, and I appreciate their clarifications. Therefore, I would like to maintain my original score.

---

### Official Review · Reviewer_gu3U · 2026-03-13

**Soundness:** 2
**Presentation:** 3
**Significance:** 3
**Originality:** 3
**Overall Recommendation:** 3
**Confidence:** 4

**Summary:**

This paper proposes the Query-Adaptive Trust-Region policy Optimization (QUATRO) for the RLVR-based LLM fine-tuning. The idea is to formulate the problem into a query-conditioned trust-region problem and leverage Lagrangian dual analysis to derive the objective. During training, the algorithm first optimizes the corresponding dual variables, and uses their optimized values to calibrate the reward as well as extract the final optimized policy. The paper is generally well-written and well-motivated. The paper still seems to have some theoretical gaps when converting the trajectory-level derivation to the token-level objective. But I am willing to increase my score if the authors can address my concerns.

**Compliance With Llm Reviewing Policy:**

Affirmed.

**Final Justification:**

The authors have addressed some of my concerns. But the issue of theoretical gaps between the trajectory-level derivation and the proposed token-level algorithm still remains.

**Key Questions For Authors:**

- Regarding the theoretical gap between the trajectory-level derivation and the proposed token-level algorithm, can you provide additional verification to demonstrate its validity?
- Can the proposed method outperform newer methods like DAPO and many of its follow-up methods?

**Limitations:**

The paper does not have a specific discussion on the limitations of this work.

**Strengths And Weaknesses:**

## Strengths
* The paper is well-motivated. The analysis of the entropy collapse and its relationship with group normalization and importance clipping is interesting.
* The idea to leverage trust-region formalism and duality analysis on the RL-based LLM fine-tuning method is valuable.
* The paper is generally well-written. The interpretations made in the paper are also reasonable.

## Weaknesses
* The entire theoretical derivation in Section 3.1 is conducted on the trajectory-level, i.e., the rewards are trajectory-level rewards, and the policy $\pi(o|q)$ is defined on the entire output trajectory. However, when turning it into the final algorithm (e.g., Theorem 3.3), the authors simply use the summed token-level version (Eq.(14)). I don't think this is mathematically correct. The trajectory-level objective has a highly non-linear form, which does not equal the summed token-level objective.
* Second, it also seems to be problematic for Eq.(19). The optimal dual variable $\lambda_q^*$

 is originally solved in the trajectory-level objective (Eq. (11)). One needs to sample lots of $o$ from $\pi_{old}(o|q)$ to optimize it. But in the final algorithm in Eq. (19), the authors simply use $N$ Monte-Carlo samples from a single $o$ to optimize $\lambda_q^*$. This treatment has a noticeable gap compared to the original theoretical derivation.
* The evaluations also seem to omit strong baselines like DAPO, which also has designs to prevent entropy collapse.


[1] DAPO: An Open-Source LLM Reinforcement Learning System at Scale. NeurIPS 2025.

---

> ### Author Rebuttal · Authors · 2026-03-31
>
> We sincerely thank the reviewer for thoughtful comments. In response, we clarify the concerns raised in W1 and W2 by providing a more explicit explanation of our trajectory-level formulation. Following the reviewer’s helpful suggestion, we also include additional comparisons with recent methods, DAPO and CISPO.
>
> ---
>
> __(W1, Q1) Trajectory-level vs. Token-level formulation__
>
> __A.__ We thank the reviewer for pointing out this ambiguity. We clarify that our derivation is consistently at the __trajectory level__. Tokens within a trajectory are denoted using the subscript $t$ (described in Sec. 2.1), while the superscript $o^{(i)}$ refers to the $i$-th trajectory out of $N$ sampled trajectories for a given query (i.e., $N$ outputs per query). We acknowledge that this is not explicitly stated when it is first introduced in Sec. 2 (GRPO), which may have caused confusion. We will clarify this in camera-ready.
>
> ---
>
> __(W2) Dual optimization and sampling__
>
> __A.__ The reviewer’s understanding is correct and aligns with our approach. We sample
> $N$ trajectories $o^{(i)}$ for $i=1,...,N$ from $π_{old}(\cdot|q)$ and optimize at the trajectory level. In practice, due to computational and memory constraints, we do not sample a large number of trajectories. The estimator gets more accurate with a larger $N$ in theory, but our empirical results suggest that a relatively small value like $N=8$ is already sufficient to achieve stable and effective performance. We will make this practical consideration clearer in the revision.
>
> ---
>
> __(W3, Q2) Comparison with recent methods (DAPO, CISPO)__
>
> __A.__ We thank the reviewer for this helpful suggestion. Following the reviewer’s recommendation, we additionally compare our method with DAPO and CISPO in the table below.
>
> |Method|Entropy|Pass@1|Pass@2|Pass@4|Pass@8|Pass@16|Pass@32|Pass@64|Pass@128|Pass@256|
> |-|-|-|-|-|-|-|-|-|-|-|
> |GSPO|0.05|32.15|37.56|42.36|46.53|50.13|53.24|55.92|58.29|60.85|
> |DAPO|0.07|31.85|37.05|41.56|45.49|48.92|51.96|54.90|58.45|63.02|
> |CISPO|0.05|31.60|36.91|41.64|45.81|49.57|53.08|56.64|60.46|64.32|
> |Ours ${\small \delta=0.1}$|0.14|**32.63**|**38.26**|**42.90**|46.75|50.23|53.63|57.12|60.61|63.65|
> |Ours ${\small \delta=0.01}$|0.20|31.02|37.12|42.45|**47.04**|51.02|54.67|58.11|61.75|65.79|
> |Ours ${\small \delta=0.001}$|**0.23**|29.73|36.18|41.91|46.96|**51.52**|**55.82**|**59.96**|**64.10**|**68.68**|
>
> We observe that DAPO and CISPO achieve strong performance with improved entropy, which supports the reviewer’s point that mitigating entropy collapse is important. Our method slightly outperforms these baselines in pass@k, with a __substantially higher entropy__, implying that our method effectively alleviates the policy concentration problem without sacrificing task performance. This is particularly meaningful, since excessive concentration makes the policy overcommit to a narrow set of outputs, reducing effective exploration and limiting the chance of discovering successful candidates across repeated samples. Our approach successfully controls policy updates in a principled way, preventing this premature concentration while still allowing steady improvement in solution quality. To sum up, we claim that our method not only achieves competitive final accuracy, but also better balances exploration-exploitation. We will clarify this interpretation in the revision.
>
> ---
>
> __(L) Limitations section__
>
> __A.__ We thank the reviewer for this helpful suggestion and will add a limitations section in the revised version. In particular, our method relies on the KL constraint parameter $\delta$, which determines how far the updated policy is allowed to move from the reference policy. This gives a principled way to control the trade-off between exploitation and exploration, but the question of how to select $\delta$ automatically, or to better understand which values work well in practice, deserves further investigation. We will clarify these limitations in the revision.

---

> > ### Author Rebuttal · Reviewer_gu3U · 2026-04-03
> >
> > I'd like to thank the authors for adding additional comparisons to other baselines.
> >
> > However, I still believe there are significant gaps between theory and practice in the proposed method. The summed token-level formulation is not equivalent to the trajectory-level formulation. For trajectory-level formulation, the sequence is modelled as a joint distribution, while for the summed token-level version, you are implicitly assuming conditional independence among tokens. These two are not consistent. For W2, the gap is even larger. Hence, I don't think the authors' responses have addressed my concern. I will keep my original score.

---

> > > ### Author Response · Authors · 2026-04-03
> > >
> > > We thank the reviewer for the prompt response. We would like to clarify the reviewer’s misunderstanding regarding the relationship between trajectory-level and token-level formulations. **Our method is defined entirely at the trajectory level**, and all quantities (objective, advantage, and likelihood ratio) are computed per trajectory.
> > >
> > > The appearance of token-wise summation arises only from the autoregressive factorization:
> > > $$\log \pi_\theta(o \mid q) = \sum_t \log \pi_\theta(o_t \mid q, o_{<t})$$
> > > which is an exact identity given by the chain rule, not an approximation or a separate token-level formulation.
> > >
> > > Importantly, this does not introduce any conditional independence assumption, as each term is conditioned on the full prefix $o_{<t}$. Therefore, the token-level summation is simply a decomposition of the trajectory-level likelihood, and the objective itself remains defined and optimized at the trajectory level.
> > >
> > > ---
> > >
> > > ### On the likelihood estimation
> > > Our implementation uses the **exact sequence-level likelihood ratio**, computed through the standard autoregressive factorization.
> > >
> > > Given the prompt (query) $q$ and the $i$-th trajectory (response) $o^{(i)} := \\{o_t^{(i)}\\}_{t=1}^{T}$, the sequence probability is written as
> > >
> > > $$\pi_\theta(o^{(i)} \mid q)=\prod_{t=1}^{T}\pi_\theta(o_t^{(i)}\mid q,o_{<t}^{(i)}),$$
> > >
> > > where $o_t^{(i)}$ denotes the $t$-th token of response $o^{(i)}$ (see also GSPO [1], Eq. (7)). Therefore,
> > >
> > > $$\log \frac{\pi_\theta(o^{(i)}\mid q)}{\pi_{\text{old}}(o^{(i)}\mid q)}=\log \pi_\theta(o^{(i)}\mid q)-\log \pi_{\text{old}}(o^{(i)}\mid q)$$
> > >
> > > $$=\log \prod_{t=1}^{T}\pi_\theta(o_t^{(i)}\mid q,o_{<t}^{(i)})
> > > -\log \prod_{t=1}^{T}\pi_{\text{old}}(o_t^{(i)}\mid q,o_{<t}^{(i)}),$$
> > > which yields
> > >
> > > $$=
> > > \sum_{t=1}^{T}\log \pi_\theta(o_t^{(i)}\mid q,o_{<t}^{(i)})-\sum_{t=1}^{T}\log \pi_{\text{old}}(o_t^{(i)}\mid q,o_{<t}^{(i)})
> > > $$
> > >
> > > $$=\sum_{t=1}^{T}\log\frac{\pi_\theta(o_t^{(i)}\mid q,o_{<t}^{(i)})}{\pi_{\text{old}}(o_t^{(i)}\mid q,o_{<t}^{(i)})}.
> > > $$
> > >
> > > Thus, the quantity we compute in implementation,
> > > $$
> > > \sum_{t=1}^{T}
> > > \log
> > > \frac{\pi_\theta(o_t^{(i)}\mid q,o_{<t}^{(i)})}
> > > {\pi_{\text{old}}(o_t^{(i)}\mid q,o_{<t}^{(i)})},
> > > $$
> > > is exactly equal to the sequence-level log-likelihood ratio
> > > $$
> > > \log \frac{\pi_\theta(o^{(i)}\mid q)}{\pi_{\text{old}}(o^{(i)}\mid q)}.
> > > $$
> > >
> > > We believe the source of confusion is that this point was not stated explicitly enough in the current version, and we will clarify it in our new manuscript.
> > >
> > > ---
> > >
> > > ### On W2.
> > >
> > > We agree that Eq. (19) corresponds to a Monte Carlo approximation of a trajectory-level expectation, and in principle, a larger number of rollouts $N$ leads to a more accurate estimate. However, in our method, we observe that even a relatively small number of samples ($N=8$) is sufficient in practice.
> > >
> > > We further provide an intuitive interpretation of the $\lambda$ computation. As illustrated in **Fig. 2** and **App. C.1**, the current $\lambda$ computation makes the effective advantage $A$ (eq. (14), (15)) depend on how many positive rollouts are present among the $N$ sampled trajectories.
> > >
> > >
> > > As shown in eq. (15) below:
> > >
> > > $$
> > > A^i_q = \frac{R(o^{(i)} \mid q) - \mu^\star_q}{\lambda_q^\star} -1.
> > > $$
> > >
> > > When a larger fraction of rollouts yields positive rewards, the optimized parameter $\lambda^\star_q$ increases. Consequently, the advantage assigned to each positive sample, $A^i_q$, is reduced.
> > >
> > > Conversely, when positive rollouts are rare, the optimized parameter $\lambda^\star_q$ decreases, resulting in a larger assigned advantage $A^i_q$.
> > > This means that the method adaptively modulates the update magnitude according to the reward distribution. In particular, when positive samples are scarce, it places stronger weight on them, which appears to be a reasonable mechanism for amplifying informative signals while avoiding unnecessarily large updates when positive outcomes are already abundant. This also helps explain why using only $N$ rollout samples is sufficient in practice: even with a limited number of samples, the method can still adapt its update strength based on the observed reward composition and extract useful training signals effectively.

---

### Decision · Program_Chairs · 2026-04-30

**Decision:**

Accept (regular)

**Comment:**

This paper proposes a query-adaptive trust-region optimization framework for RL-based LLM fine-tuning. The work is well-motivated, addressing known limitations of GRPO-style methods such as instability under policy staleness and reliance on heuristic clipping. The core idea is technically meaningful, and reviewers find the resulting formulation both interpretable and practically relevant. Empirically, the method shows consistent improvements and improved stability on mathematical reasoning benchmarks, with additional analyses supporting its robustness.

A central concern raised during review was a potential mismatch between the trajectory-level theoretical derivation and the token-level implementation. Based on the rebuttal and further examination, this issue appears to be largely due to clarity and notation in the presentation, which could be improved in the final version. A related point regarding the Monte Carlo approximation used for dual variable estimation is valid and should be better discussed. The empirical evaluation is primarily focused on mathematical reasoning tasks and a limited set of model families, leaving some uncertainty about generalization. Additionally, comparisons to some recent baselines could be strengthened.

Overall, while the novelty is moderate and the experimental scope could be broader, the paper provides a principled and practically useful contribution to RL-based LLM fine-tuning. The method is technically sound, empirically supported, and addresses an important problem.